# Online conversion of reconstructed neural morphologies into standardized SWC format

Ketan Mehta [1,3], Bengt Ljungquist [1,3], James Ogden[1], Sumit Nanda[1], Ruben G. Ascoli[1], Lydia Ng[2] & Giorgio A. Ascoli [1] ✉

Digital reconstructions provide an accurate and reliable way to store, share, model, quantify, and analyze neural morphology. Continuous advances in cellular labeling, tissue processing, microscopic imaging, and automated tracing catalyzed a proliferation of software applications to reconstruct neural morphology. These computer programs typically encode the data in custom file formats. The resulting format heterogeneity severely hampers the interoperability and reusability of these valuable data. Among these many alternatives, the SWC file format has emerged as a popular community choice, coalescing a rich ecosystem of related neuroinformatics resources for tracing, visualization, analysis, and simulation. This report presents a standardized specification of the SWC file format. In addition, we introduce *xyz2swc*, a free online service that converts all 26 reconstruction formats (and 72 variations) described in the scientific literature into the SWC standard. The *xyz2swc* service is available open source through a user-friendly browser interface (https://neuromorpho.org/xyz2swc/ui/) and an Application Programming Interface (API).

A digital reconstruction of a neuronal or glial (henceforth referred to as neural) cell is an information-rich numerical representation of its tree structure[1,2]. The most common type of digital reconstruction represents neural morphology as a skeletonized tracing: a series of interconnected points that capture neurite position, thickness, connectivity, and structural domain (e.g., dendrite vs. axon). By providing an accurate and reliable way to store, share, analyze, and model neural morphology, digital reconstructions have broad applicability across several subdisciplines of neuroscience. Digital reconstructions enable thorough morphometric comparisons among species, developmental stages, brain regions, cell types, and pathophysiological conditions[3]. Digital reconstructions are also instrumental in realistic electrophysiological simulations through the compartmental representation of active and passive membrane properties as well as synaptic signals[4]. At the circuit level, digital reconstructions allow the quantitative estimation of potential network connectivity[5]. Moreover, thanks to their clear distillation of morphological details, neural reconstructions are increasingly utilized in education and outreach.

Continuous advancements in microscopic imaging, cellular labeling, tissue processing, and automated tracing catalyzed a rapid growth of available data and a related proliferation of software applications pertaining to neural morphology. These include computer programs for digital tracing and visualization, such as Neurolucida[6] (MBF Bioscience), Imaris (Oxford Instruments), Amira[7] (ThermoFisher), KNOSSOS[8], Eutectics[9], SNT[10,11], NeuronJ[12], and the HBP Morphology Viewer[13], as well as for modeling, including TREES Toolbox[14], NEURON[15,16], and Genesis[17]. These software programs typically use custom digital file formats to encode and store the digital reconstruction data. This digital format heterogeneity constitutes a major impediment when it comes to interoperability and reusability of neural reconstructions. The neuroscience community increasingly recognizes the benefits of Findable, Accessible, Interoperable, and Reusable (FAIR[18]) resources[19]. Possible FAIR file format candidates for digital reconstructions of neural morphology include XML (Extensible Markup Language) by MBF Bioscience[20], NML by NeuroML[21–23], and SWC (named after its initial designers Ed Stockley, Howard Wheal, and

[1]Center for Neural Informatics, Structures & Plasticity, George Mason University, Fairfax, VA, USA. [2]Allen Institute for Brain Science, Seattle, WA, USA. [3]These authors contributed equally: Ketan Mehta, Bengt Ljungquist. ✉e-mail: ascoli@gmu.edu

Robert Cannon), which was introduced in the first public online archive of traced neuronal arbors[24].

The SWC format has emerged as the most broadly recognized and used format in the community, with a rich ecosystem of related software for tracing, visualizing, analyzing, and modeling neural morphology[25]. The SWC popularity stems in part from its information-rich, yet very concise, representation[26]. In addition, the format is open source, not controlled by a commercial entity, and readable by both humans and machines. The SWC format has also been adopted, with minor extensions, by NeuroMorpho.Org[27], to date the largest freely accessible database of neural reconstructions.

Despite its widespread usage, the SWC format still lacks community standardization. Multiple SWC variants have emerged, causing confusion among neuroscientists, as evidenced by several email exchanges on this topic on the NEURON listserv. Based on recurrent feedback by NeuroMorpho.Org users, the propagation and proliferation of differences among SWC file formats has also slowed down the development and adoption of open-source tools striving to maximize interoperability. Furthermore, accurate and reliable conversion to SWC from all other digital formats remains an open challenge. The formal specifications for most of these formats are not publicly available, thus requiring their schema to be reverse engineered. Therefore, currently existing conversion software tools[13,28–31] are restricted in scope and usability, and a "universal SWC converter" is still missing.

To mitigate these issues, we present here a standardized specification of the minimum requirements for the SWC file format. In addition, we release *xyz2swc* (RRID:SCR_023317), a publicly available online service that imports different reconstruction formats and exports them as a standardized SWC file. It supports as input 26 different formats and 72 variations thereof, effectively covering all reconstruction software programs described in the scientific literature. The *xyz2swc* service also provides the functionality to verify and correct non-compliant SWC files to ensure that they meet the standard specification. Its modular software architecture wraps together existing open-source converters wherever possible, is operating system and programming language independent, and requires no local package installation. This work aims to promote data sharing efforts and facilitate compatibility between scientific applications.

## Results

### Standardized specification for the SWC file format

An SWC file is a delimited text file that stores the digital reconstruction of a neural morphology (or specific portions thereof) as a data table with rows and columns (Fig. 1a). Every line in the text file (row) contains seven ordered data fields (columns) delimited by whitespace and represents a sample point traced along the neural tree (Fig. 1b, Supplementary Movie 1).

The first data field, the Index, is a positive integer uniquely identifying each sample point. The first sample point in the SWC file is assigned Index 1, and the Index of each subsequent sample increases sequentially (1,2,3, …). The second data field, Type, is an integer value (Table 1) defining the structural domain of the sample point. It is worth noting that, although for many years use of SWC files was practically limited to describing neurons, application to glia grew dramatically in recent times[32]. Currently, 27.5% of NeuroMorpho.Org reconstructions are glial tracings, the single largest dataset is a glia study[33], and the most represented cell type in the entire database is microglia, hence the dedicated type for glial processes. The 'custom' Type (5) is available for users to interpret at will, such as oblique dendrite[34], and all value above 7 may also be used for this purpose when needing to specify more than a single additional feature. It is important to clarify the distinction between the 'undefined' Type (0) and the 'unspecified neurite' Type (6). The former is used when the structural domain of an SWC point is not established in the reconstruction, and could correspond to any portion of the morphology, including soma or glia process. In contrast, Type 6 is used when the structural domain is certainly a neurite (dendrites or axon), but the available information is insufficient to discriminate between the exact type of neurite. This is often the case in dissociated cultures, in early developmental stages (when axons and dendrites are not fully differentiated), and in some invertebrate nervous systems where the same neurite can occasionally serve as both dendrite and axon.

The next three data fields are the X, Y, Z Cartesian coordinates of the sample point in three-dimensional space, respectively. The sixth field, Radius, equals half the thickness of the neural tree segment at the location specified by the X, Y, Z coordinates. The standard specification requires that these 4 data fields be expressed as floating-point values. Unless otherwise noted, they are typically interpreted (e.g., in NeuroMorpho.Org) as representing micrometers. The final data field is the Parent index, which defines how the sample points are connected to each other. The first point in the file must be the root and has a Parent of −1. All other points have a previously declared Index as a Parent. Note that, in a tree, every point can only have one Parent, but multiple points can have the same Parent.

The SWC digital file is a text file suffixed with a.swc file extension and encoded using the American Standard Code for Information Interchange (ASCII). ASCII is the most common character encoding format for text data in computers and includes upper- and lower-case letters A through Z, numbers 0 through 9, and basic punctuation symbols. The SWC file allows for optional header and footer sections in which each line starts with the hash symbol (#). The header section can be used to organize and store metadata information about the reconstruction data (digitization source, authors, brain region, cell type, version number, recording date, etc.). In addition, the optional footer section provides back-compatibility with extended multi-signal SWC (.eswc) files and can be used to include information such as signal intensity of the multiple imaged channels, or time-dependent changes in structure[35]. While the standardized specification for the SWC format does not dictate which information should be included in the file header and footer, nor how it should be organized, the next section offers our recommendations in those regards.

The soma in a standardized SWC can consist of a single root point or multiple points, the first of which must be a root. The single-point representation approximates the soma as a sphere of radius R that is centered at X, Y, Z. The multi-point representation approximates the soma as a sequence of nodes akin to a neurite branch.

We have made the standardized specification for the SWC file format publicly available (https://swc-specification.readthedocs.io) and initiated version management to encourage communal development (https://www.incf.org/commentaries/swc), following and promoting FAIR principles[18]. Therefore, the SWC format is now Findable, as metadata may be stored together with the data in the file, using hash tags as described above, including a unique identifier. Moreover, its versioned specifications are posted in permanent repositories regularly indexed by major internet search engines. It is Accessible, as the already widely utilized format is both human and machine readable, formally standardized, and open for community-driven improvement. The format is Interoperable, as it uses vocabularies widely used by NeuroMorpho.Org, which have been created and harmonized in collaboration with over 1000 labs worldwide, and also allows for cross-linking of resource identifiers as part of the metadata. In addition, it is compatible with a large ecosystem of visualization, analysis, and modeling software programs. Lastly, it is Reusable, as it allows metadata referencing to publications, contributors, experimental protocols, and a variety of other descriptor labels; its clear open-source definition also readily allow broad dissemination.

The first version (SWC v1.0.0), described above, was discussed and approved by the BRAIN Initiative Cell Census Network (BICCN) Anatomy and Morphology Working Group in 2022. Further

**a**

| Index | Type | X | Y | Z | R | Parent |
|---|---|---|---|---|---|---|
| *Description* Sample identifier | Structure identifier | 3D Cartesian coordinate | | | Radius of segment at sample point | Connectivity sample identifier |
| *Value* Sequential positive integer | Non-negative integer | Floating point | | | Positive floating point | Positive integer or -1 |
| 1 | 1 | 0.0 | 0.0 | 0.0 | 5.0 | -1 |
| 2 | 3 | 0.0 | 20.0 | 0.0 | 1.8 | 1 |
| 3 | 3 | -5.0 | 40.0 | 10.0 | 1.4 | 2 |
| 4 | 3 | -5.0 | 55.0 | 10.0 | 1.2 | 3 |
| 5 | 3 | -5.0 | 60.0 | 20.0 | 1.0 | 4 |
| 6 | 3 | 0.0 | 70.0 | 30.0 | 1.0 | 5 |
| 7 | 3 | 0.0 | 70.0 | 0.0 | 0.8 | 4 |
| 8 | 3 | 10.0 | 30.0 | 10.0 | 1.4 | 2 |
| 9 | 3 | 15.0 | 40.0 | 10.0 | 1.4 | 8 |
| 10 | 3 | 30.0 | 55.0 | 0.0 | 1.0 | 9 |
| 11 | 3 | 20.0 | 45.0 | 20.0 | 1.4 | 9 |
| 12 | 2 | 5.0 | -20.0 | 20.0 | 1.0 | 1 |
| 13 | 2 | 12.0 | -20.0 | 22.0 | 1.0 | 12 |
| 14 | 2 | 0.0 | -30.0 | 30.0 | 0.8 | 12 |

**b**

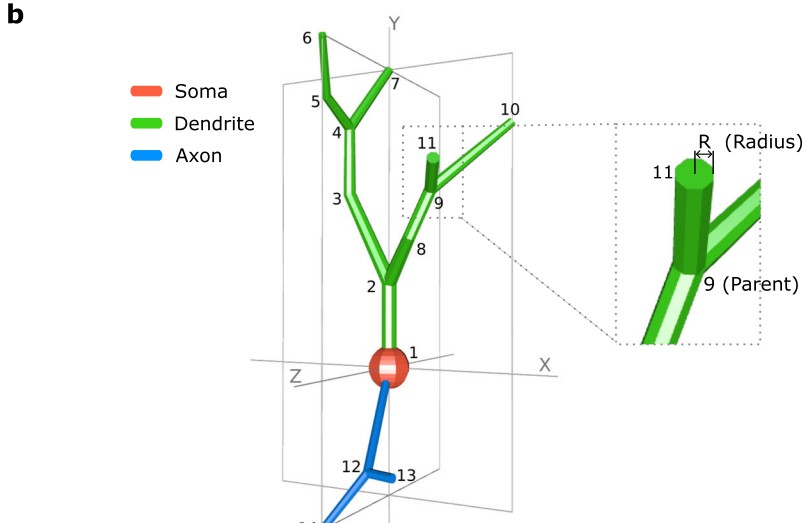

Soma
Dendrite
Axon

**Fig. 1 | SWC-formatted arbor morphology. a** Standardized SWC v1.0.0 representation for an extremely simplified neuronal structure with a total of 14 traced sample points. Each sample point in the SWC reconstruction file is characterized by seven data field values ordered by column. **b** Corresponding digital reconstruction of the tree structure with soma (red), dendrites (green), and axon (blue).

development of the standard will be coordinated by a governing board with 5 representatives: (1) NeuroMorpho.Org representative, (2) Allen Institute representative, (3) cellular neuroanatomy expert, (4) computational modeling expert, and (5) INCF advisor. Of these, the cellular neuroanatomy and computational modeling experts are appointed by election, while the others are appointed by the respective organization. Community input is continuously considered through the GitHub repository issue tracking system (https://github.com/INCF/swc-specification).

**Recommendations for optional inclusion of ancillary information in SWC files**

As mentioned above, individual researchers may choose to specify additional details in the header or footer of the SWC file (Supplementary Fig. 1). In particular, the header is most commonly employed to relay metadata information, as originally proposed[24]. Since many of the same metadata elements are frequently employed across studies, we provide here our suggestions based on the most common annotations in use at NeuroMorpho.Org[36]. Specifically, one or more lines starting with the "#" sign should convey the following information, if known: the name of the author(s), dataset, or lab of origin (#contributor); DOI or full bibliographic citation for document describing data (#reference); the animal species and strain or genotype (#creature), sex (#sex), age (#age), and weight (#weight); the anatomical region of the cell body (#region) and the cell type or identifying features (#class); the experimental group (#condition); the labeling or staining (#label); the slicing direction and thickness (#slicing); the objective type and magnification (#microscopy); the physical units (#coordinate), reference frame (#brainspace), and the tracing software (#original_source). It is important to recognize that the above list cannot capture, nor is it always applicable to, all essential details of each neuroanatomy study. Moreover, to truly standardize metadata it would be necessary to define not only the required fields, but also a set of controlled vocabularies to describe the corresponding details[37]. Recent developments in machine learning can greatly facilitate this process[38].

**Table 1 | Specified integer values for the second data field, Type, in the SWC v1.0.0 format**

| Type | Structural domain (Description) |
|---|---|
| 0 | Undefined (structure type unknown or unspecified) |
| 1 | Soma |
| 2 | Axon |
| 3 | Basal dendrite |
| 4 | Apical dendrite |
| 5 | Custom (user defined element) |
| 6 | Unspecified neurite (axon or dendrite) |
| 7 | Glia process |
| >7 | Custom (additional user defined element) |

A second opportunity for increasing the applicability of SWC files is to utilize the footer to convey information regarding synaptic connectivity. Here we adopt the format recently proposed by the fly electron microscopy community[39,40]. Accordingly, the synaptic connectivity in the SWC file footer should begin with a "#start synapse" line and finish with an "#end synapse" line. These delimiters are useful to avoid accidentally reading as synapses other cellular information that users might want to include in the footer, such as organelle distributions or temporal branch dynamics[35]. In between, each line should describe a synaptic contact with a "#" sign followed by 9 fields: (i) a unique ID for the detected synapse; (ii–iv) the x, y, and z synapse position in coordinate space; (v) the node in the SWC file closest to the synapse; (vi) a binary assignment with 0 indicating an output synapse and 1 indicating an input synapse; (vii) the neurite structural domain, e.g., axon vs. dendrite, analogous to the second data field (type) in the SWC data file; (viii) a unique identifier for the partnering neuron; and (ix) the putative neurotransmitter. Prior to the first synapse data line, a line (always starting with "#") should list the meaning of these fields.

The above information could be parsed automatically by suitable connectomics analysis platforms[41,42]. At the same time, because traditional SWC readers interpret the "#" as a comment indicator, the proposed header and footer formats ensure continuous back-compatibility with legacy software. Examples of standard SWC files with metadata and synaptic connectivity information included as described in this section are provided as Supplementary Data 1 and Supplementary Data 2, respectively.

### The xyz2swc conversion and standardization service

We have developed the *xyz2swc* conversion software that allows convenient conversion of neural digital reconstructions from any common format into the SWC standard format. The tool is deployed as a publicly available online service and has two main functionalities: (1) import files of different reconstruction formats and export them as standardized SWC files; and (2) import existing SWC files for verification (and correction if needed) to ensure they meet the standardized specification of this format.

The *xyz2swc* service is freely accessible through any common internet browser through a user-friendly web-based graphical interface at https://neuromorpho.org/xyz2swc/ui/ (Fig. 2a, Supplementary Movie 2). To convert their digital tracings, users simply upload the reconstruction files (either individually or as a zipped archive) and select the "Convert/Standardize" option. The service automatically detects the format of the uploaded files, performs the data conversion, and provides the converted standardized SWC files for download. Using the application does not require prior knowledge of the format specification of the original reconstruction files nor of the SWC format. For imported files that are already in SWC format, the service also provides a "Check" only option, which verifies (without converting) if the file meets standard specification and returns a summary log of any non-standard formatting issues.

We have developed a representational state transfer (REST) Application Programming Interface (API) for *xyz2swc* that facilitates software interaction and programmatic use of the service. The API allows convenient public access to all conversion and standardization capabilities from almost any modern programming language. Using the API does not require a software license, nor does it require the user to install any part of the *xyz2swc* service onto their local computer. A description of the API along with a list of possible commands is available at https://neuromorpho.org/xyz2swc/docs.

The *xyz2swc* service is deployed as a Docker container (Fig. 2b) for the benefit of version control, rapid updates, lower maintenance downtime, and effective scaling on potential periods of higher load. In addition, the published Docker image (https://hub.docker.com/r/neuromorpho/xyz2swc) contains the latest stable version of the source code, libraries, modules, and all other dependencies needed to install and run the service locally, e.g., on a private server if desired.

### Mass validation of SWC conversion robustness using NeuroMorpho.Org data

The *xyz2swc* service supports SWC conversion of 26 different file formats and over 72 format variations thereof (see Methods and Supplementary Table 1). These include reconstructions generated by popular open-source tracing software (e.g., SNT, KNOSSOS, NeuronJ), by commercial closed-license programs (e.g., Neurolucida, Imaris, Amira), and by morphological and electrophysiological modeling applications (TREES toolbox, NEURON, Genesis, PSICS[43]). To our knowledge, *xyz2swc* covers all neural reconstruction formats described in the peer-reviewed literature or on the internet including relatively newer formats developed for open data sharing (NeuroML, SWC+) and legacy formats originally designed by individual labs (Arbor[44], Nevin[45]).

These file formats vary in the manner they represent neural morphology. For example, Neurolucida uses a hierarchical tree structure, TREES toolbox represents arbors as a directed adjacency matrix, and Amira adopts a lattice point representation. The formats are also diverse in their digital encoding of the reconstruction data, such as an ASCII text file in NeuronJ.ndf files and a compressed binary file in SNT.traces files. Furthermore, there exist variations within the file structures of the same format across tracing software programs. For instance, the formatting differs between HOC files generated by NEURON, Imaris, and Eutectic.

The *xyz2swc* support of such multifarious inputs derives from its modular integration of original novel code with custom modifications of existing open-source programs written in several languages (Fig. 3). This design allows for adding or updating individual software components to accept new formats or variations, or to include further features, without impacting the overall service operation.

In order to test *xyz2swc*, we performed an automated mass validation using all digital reconstructions available on the NeuroMorpho.Org (version 8.4) repository. This online database is to date the largest collection of publicly accessible neural tracings. While all morphologies are converted into SWC when ingested into NeuroMorpho.Org[35], this resource also makes the original data files available for download as provided by the contributors in the format in which they were initially collected. Of 232,029 digital reconstructions, 143,131 encompassed 26 reconstruction formats and over 72 variations from 24 software programs (Table 2). The other 88,898 files were native SWC files generated from 36 different software programs.

*Xyz2swc* converted 142,605 of non-SWC files from NeuroMorpho.Org v.8.4, corresponding to an overall success rate of 99.63% (a detailed report of the mass validation results for all files is included in Supplementary Table 1). These data corroborate the robustness of

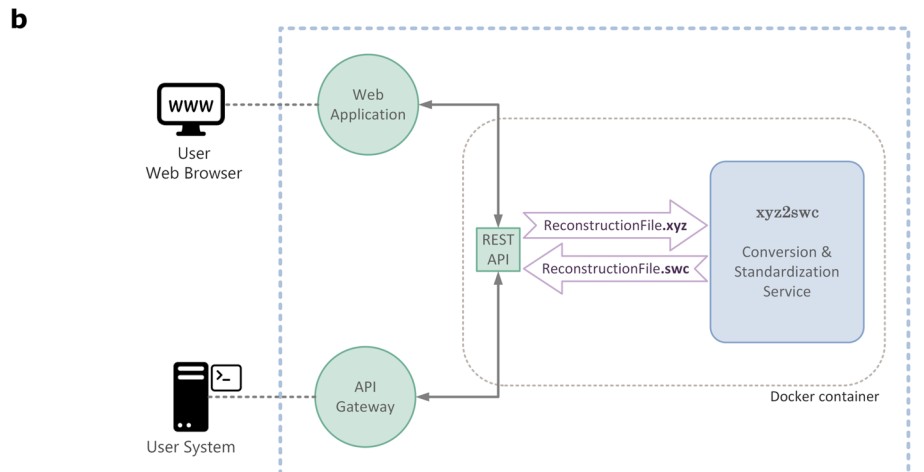

**Fig. 2 | The *xyz2swc* web service. a** A simple graphical interface provides user-friendly access to the underlying API to convert any format into SWC standard. **b** The system is deployed as a Docker container accessed by a Nginx web server, providing reverse proxy and load balancing.

*xyz2swc* as a universal converter. The files that failed to convert had a format variation that differed from the documented version in ways that could not be reliably interpreted.

When the imported format lacks information pertaining to one of the SWC data fields, *xyz2swc* automatically inserts default values. For example, the NeuronJ NDF format does not store the thickness and depth of each branch, capturing the arbor as a two-dimensional linear projection. The resultant SWC file is given a uniform Radius of 0.5 and Z coordinates of 0 (Fig. 4a and Table 3). Similarly, the TREES Toolbox MTR format does not specify the structural domain (e.g., axon, dendrites, and soma). The converted SWC files are thus assigned Type 0 (undefined) for all sample points (Fig. 4b). In all cases, visual checks confirm the physical integrity of the neural tree before and after conversion. Conversely, certain formats include additional features that are not supported by SWC. For example, Neurolucida DAT, ASC, and XML formats provide options to annotate subcellular structures, such as spines, varicosities, and puncta, anatomical boundaries, and user-defined textual markers. In these cases, *xyz2swc* converts all data supported by the SWC specification while omitting the rest.

As digital reconstructions become more prevalent, a key consideration when sharing and archiving tracing data is their file size. For the validated formats, Table 2 compares the ratio between the average file size of the converted SWC files and that of the original formats. The results indicate that when encoding neuron data there is an inherent trade-off between feature support and file size, with the SWC format providing a significant storage gain when compared to feature-rich

formats such as XML, NML and IMS. The only formats having a smaller storage footprint than SWC are compressed binary files (NRX, NMX, MAT, MTR) and the simple two-dimensional NDF format. However, we can achieve better or similar storage gains to these compressed formats by simply gzip[46] compressing their equivalent SWCs (compressed file size ratio listed within parenthesis in Table 2).

Of note, representing the soma as a contour tracing of the cell body perimeter or a series of contours approximating the somatic surface is not consistent with SWC v1.0.0. The *xyz2swc* standardization module automatically detects soma contours in the imported file and converts them into an equivalent standardized representation (Fig. 5). Specifically, the program measures the curvature of all soma sections in the morphological representation (Fig. 5a). If the curvature angle is obtuse ($\theta \geq 90°$), the soma section is interpreted as a series of stacked frustums and retained as is. If the curvature angle is acute ($\theta < 90°$), the soma section is deemed to be a contour and replaced. A single soma contour (Fig. 5b) is replaced by a single root point whose X, Y, Z coordinates correspond to the center of the contour, i.e., the average of all contour points. The Radius is computed as the average distance of each contour point from this center. If multiple contours are present (Fig. 5c), they are transformed into a series of points, each following the above procedure.

## SWC standardization robustness
The *xyz2swc* service provides support for standardizing all SWC format variations represented in the original files of NeuroMorpho.Org v.8.4.

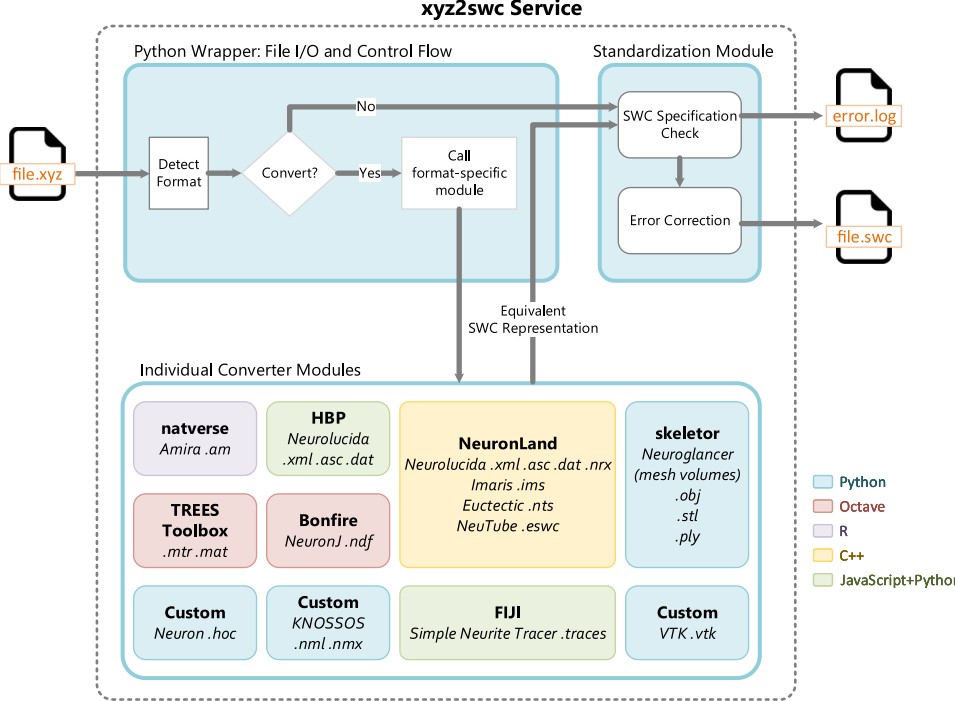

**Fig. 3 | The backend workflow of the *xyz2swc* service.** Consists of multiple independent modules written in different programming languages, each contributing different functionalities, all wrapped together with Python.

Specifically, *xyz2swc* performs a series of checks to evaluate if the imported SWC file meets the standardized specification (Table 3). These checks include common formatting errors such as parent samples being referred to before they are defined, negative radius values, and contour soma representations. Other checks take care of non-standard formatting styles used by popular software programs. For example, SWC files generated by NeuronStudio use Type values of 5 and 6 to indicate respectively bifurcations and terminals, inconsistent with the standardized specification. A SWC file is considered standardized only if it passes all checks. If not, *xyz2swc* can correct the detected errors

**Table 2 | Mass validation for the most popular reconstruction formats on NeuroMorpho.Org**

| Software application | File format | File count | Avg. File-Size Ratio (Compressed SWC) | Conversion rate (%) |
|---|---|---|---|---|
| Neurolucida | .dat | 59,268 | 0.79 | 100.00 |
| Neurolucida | .asc | 19,673 | 0.69 | 99.86 |
| Neurolucida | .nrx | 2810 | 1.88 (0.48) | 100.00 |
| Neurolucida | .xml | 96 | 0.41 | 100.00 |
| Amira | .am | 16,536 | 0.84 | 99.21 |
| NeuronJ | .ndf | 7844 | 4.14 | 100.00 |
| TREES Toolbox | .mat | 962 | 1.58 (0.84) | 100.00 |
| TREES Toolbox | .mtr | 260 | 3.43 (1.11) | 100.00 |
| KNOSSOS | .nml | 3261 | 0.31 | 100.00 |
| KNOSSOS | .xml | 837 | 0.18 | 100.00 |
| PyKNOSSOS | .nmx | 1021 | 2.81 (0.66) | 100.00 |
| SNT | .traces | 13,146 | 0.84 | 100.00 |
| Imaris | .ims | 12,020 | 0.02 | 99.81 |
| Imaris | .hoc | 3049 | 0.97 | 99.86 |
| Eutectic | .nts | 563 | 1.05 | 100.00 |
| NeuronStudio | .eswc | 118 | 0.59 | 100.00 |
| Others | various | 1667 | 0.45 | 79.54 |

(Table 3). In particular, the "Check" option only outputs a log file of any detected errors, allowing users to just verify if their file meets the standardized specification without correction. Alternatively, the "Convert/Standardize" option exports the data into the specified standard.

Certain errors cannot be corrected, for example when an entire column of data is missing, or the supposedly SWC-formatted file is not ASCII-encoded. These errors often indicate file corruption or a non-SWC files incorrectly suffixed with a.swc extension. In those cases, *xyz2swc* still outputs the log file to aid users in handling the identified issues. The standardization success rate of the tested data was 100%. In other words, all 88,898 SWC-formatted original files in NeuroMorpho.org v.8.4 either already met the SWC v1.0.0 standard or were successfully standardized by *xyz2swc*.

Note that SWC files that nominally meet standardized specification can still represent erroneous or inexact reconstructions of the original neural structure. Some of these errors may be evident on visual inspection, especially when corresponding to mistakenly connected branches. Other software programs provide functionality to detect and correct those issues[30,47].

## Discussion

We provide here a standard for the widely used SWC file format, as well as a software tool (publicly accessible both via a web interface and programmatically) enabling the conversion of all known digital reconstruction formats and variations thereof into standardized SWC data. The proliferation of neural reconstruction formats arose by a lack of clear documentation and constitutes an impediment to effective reuse of digital reconstructions. The public standard specification and open-source conversion service are together an effort to encourage communal development, achieve greater interoperability, and promote data-driven science following FAIR principles. This is similar to analogous approaches in related fields, such as the Open Microscopy Environment for managing differences between microscope vendors[48] and the Systems Biology Markup Language for unambiguously exchanging biological model descriptions[49]. The modular architecture of *xyz2swc* allows for community-guided functional extension and

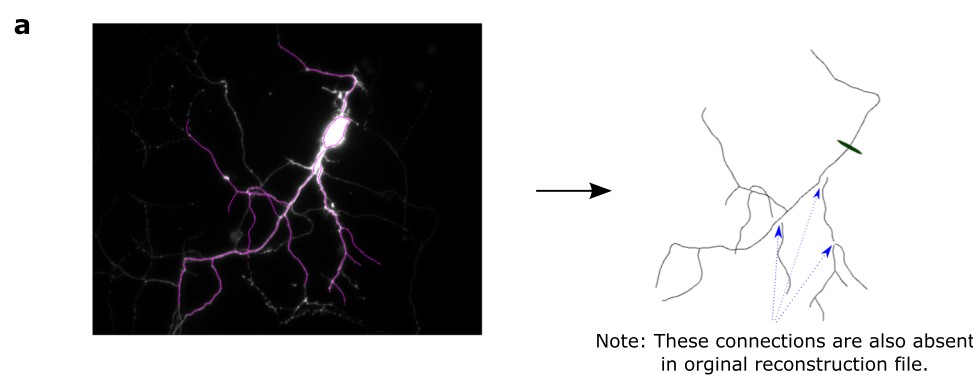

**a**

Note: These connections are also absent
in orginal reconstruction file.

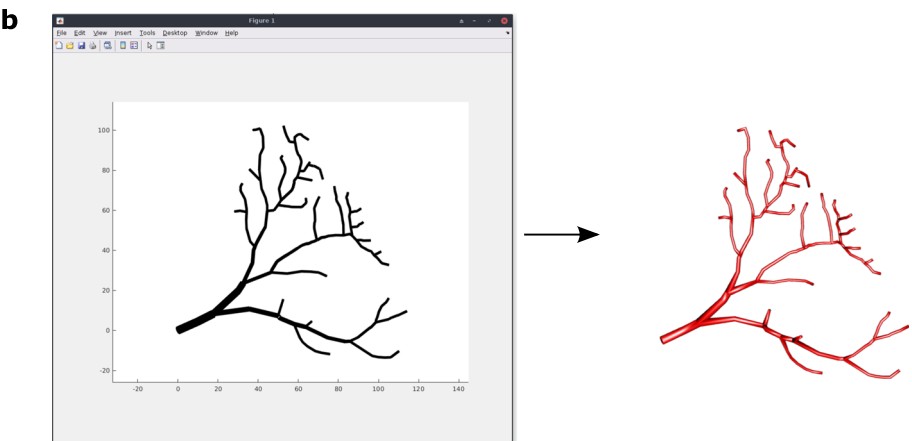

**b**

**Fig. 4 | Format conversion accurately preserves neural structure. a** A NeuronJ NDF reconstruction converted to SWC. **b** A TREES Toolbox MTR reconstruction converted to SWC.

**Table 3 | List of specification checks and corresponding error correction (in the order of execution) to ensure SWC standardization**

| Check | Action/Correction |
|---|---|
| Missing Field | If the SWC points matrix does not have seven columns, then return an error. All further checks are omitted. |
| Number of Lines | - Generate an error if no samples are detected. All further checks are omitted.<br>- If fewer than 20 lines, generate a warning to check file integrity. |
| Number of soma Samples | Generate warning if no soma samples detected. |
| Invalid Parent | If the Parent points to an Index value that does not exist, then make the sample with the invalid Parent a root point, and generate a warning to check file integrity. |
| Index/Parent Integer | If Index and/or Parent are float-formatted integer (e.g., "1.00"), format them as integers. If they are non-integer values (e.g., "1.34") or non-numerical entries (e.g., "abc"), generate an error. |
| XYZ Double | Ensure X, Y, and Z coordinates are float/double values. Any NaN or NA values detected in the ASCII text file are treated as 0.0. Generate a warning to check file integrity, and add a footer to the file to note inserted values. |
| Radius Positive Double | - Ensure sample Radius is a double/float value.<br>- If radius is negative, zero, NaN, or NA, then set to 0.5. Generate a warning to check file integrity, and add a footer to the file to note inserted values. |
| Non-Standard Type | - If Type is float-formatted integer, format as integer.<br>- If it is non-integer value or non-numerical entry, change to Type 0 indicating 'undefined'.<br>- If bifurcation and terminal points have non-standard Types, set them to that of parent. |
| Sequential Index | If the Index values are not in sequential order (starting from 1), then sort and reset Index and Parent numbering. |
| Sorted Order | - If parent samples are referred to before being defined, then sort and reset Index and Parent numbering.<br>- Sort indices to ensure that the first sample in the file is a root point. If no sample point is a root, generate an error. |
| Soma Contours | Detect soma contour(s), and replace each with a single point. |

integration of additional format variations when needed[50]. The mass validation results demonstrate the broad applicability of both the conversion service and standardized specification, effectively encompassing all publicly available neural reconstruction formats.

A strength of SWC is its simplicity, with the concise representation of the neural arbor branching skeleton that is sufficient for most research applications[26]. While more specialized feature-rich formats may be useful in some cases, they often present an obstacle to usability

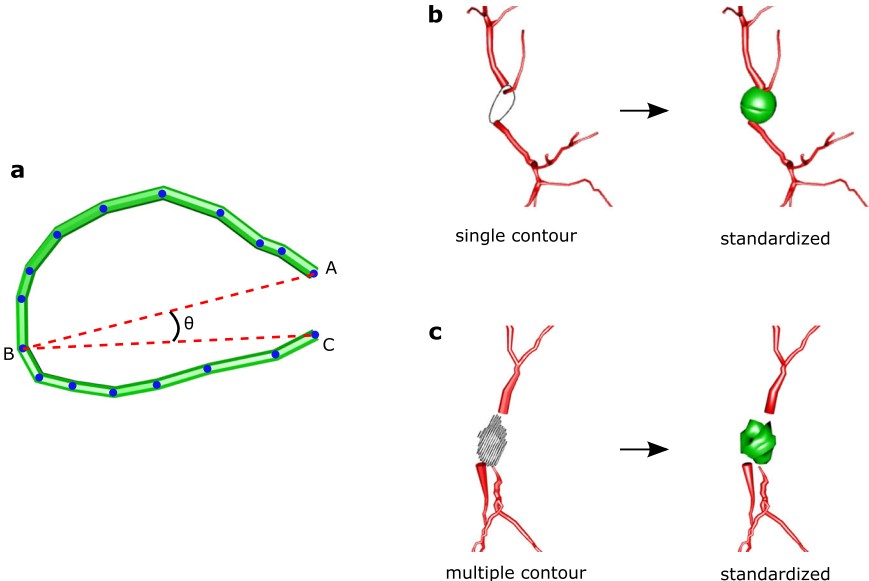

**Fig. 5 | Soma conversion. a** The curvature angle $\theta$ of a series of soma sample points is used to determine whether the section is a contour. **b** Conversion of a Neurolucida ASC file with a single soma contour into its SWC v1.0.0 representation. **c** Conversion of a Neurolucida DAT file with multiple soma contours into SWC v1.0.0.

and maintainability. For example, the Neurolucida's XML format[20], while abiding by FAIR principles, is not backwards compatible with the more commonly adopted DAT and ASC formats from the same software, nor with the XML specification of NeuroML[22]. This is also why, of the several variants spawned from the original SWC, we chose to standardize the format adopted by NeuroMorpho.Org. This format extends the original description[24] as necessary to designate unspecified neurites and glial processes, while allowing back-compatible extensions, e.g., for the inclusion of subcellular and time-lapse data as in ESWC files[35] or of synaptic connectivity as proposed in the fly electron microscopy community[39,40] and briefly outlined in this paper. In contrast, other variants such as SWC+[13], while useful to specify layer boundaries and other details, have yet to gain substantial traction in the broader research community, possibly because experimental anatomists may find the adopted XML formalism less intuitively accessible. Furthermore, not all studies require all the features encoded in the more complex formats. Therefore, the ability to rely on a common, standardized SWC lingua franca ensures that the reconstruction data remains compatible, accessible, and reusable for future analyses.

Recent technological advancements in automated tracing, tissue preparation techniques, and imaging modalities make it now possible to reconstruct neural morphology at unprecedented rates[51,52]. The parallel, rapid growth of computational neuroscience in general has resulted in an ever-increasing emphasis on the role of digital reconstructions in characterizing network connectivity, modeling brain circuitry, and investigating the structure-function relationship in the nervous system. This trend is most apparent when noting the explicit involvement of big-data neuroscience programs in collecting digital reconstructions of neural morphology[53,54], including the US BRAIN Initiative (braininitiative.nih.gov) and the EU Human Brain Project (humanbrainproject.eu), and the corresponding growth of large scale databases such as NeuroMorpho.Org, MouseLight[55], and FlyCircuit[56]. These collaborative endeavors require the systematic organization and FAIR standardization of reconstruction data[19]. While it is unlikely that all neuroscientists will ever agree to trace their neural morphologies in a single format, universal conversion ensure efficient reusability of all available data[57].

## Methods

The *xyz2swc* RESTful API is implemented using Python and FastAPI. The API definition allows the following methods: (1) Upload file(s); (2) Convert uploaded file(s); (3) Standardize uploaded file(s); (4) Get converted file(s) as a zip archive; (5) Get conversion and standardization log(s) as a zip archive; and (6) Clear uploaded files. User API calls are authenticated via a key that is provided when initiating a session. Uploaded files are automatically cleared out after 24 h. A full description of the API is available together with the source code and its further documentation at https://neuromorpho.org/xyz2swc/docs. The web user interface directly utilizes the underlying API, and was implemented using Bootstrap—a CSS, HTML, and JavaScript components library. The *xyz2swc* service is deployed as a Docker container using Ubuntu Linux 18.04 LTS as the operating system on a virtual machine hosted at the data center of George Mason University's Office of Research Computing. A Nginx web server provides public access to the Docker container for reverse proxy and load balancing.

To comprehensively cover all file formats and their variations, *xyz2swc* adopts a modular programming architecture integrating multiple independent open-source converters[11,13,14,28,29,31,58,59] written in different programming languages (Table 4). When prior converters were inadequate or did not exist, we developed custom scripts by reverse engineering the schema behind the formats. All modules are embedded in the overall service using a Python wrapper. The wrapper automatically detects the format of the input file and calls the appropriate converter module. The converter module then decodes the file structure, extracts the reconstruction data, and translates this information into SWC.

The Python wrapper provides convenient top-level control for upgrading, adding, or removing individual modules without impacting the operation of other modules within the service. Support for a new morphology format can be integrated into the existing *xyz2swc* service by creating a new Python module and simply importing it via the wrapper (source code[60] along with example available at https://github.com/neuromorpho/xyz2swc). Of note, it is not currently possible to run *xyz2swc* in reverse to convert SWC files to other skeleton formats. However, that functionality is provided by the NeuronLand converter for the formats supported by that module, whose source code is also released with this paper.

**Table 4 | List of supported file formats and variations thereof, with their converter modules and corresponding programming language**

| Software application | File format | Test data source | Converter module(s) | Programming language(s) | No. of variations supported |
|---|---|---|---|---|---|
| Amira[7, 61] (ThermoFisher, RRID:SCR_007353) | .am | Neuromorpho.Org | natverse[28] | R | 3 |
| Arbor[44] | .swc | Neuromorpho.Org | NeuronLand | C++ | 1 |
| ESWC[35] | .eswc | Neuromorpho.Org | Custom | Python | 2 |
| Eutectics[9] | .nts | Neuromorpho.Org | NeuronLand; Custom | C++; Python | 4 |
| Genesis[17] | .p | senselab.med.yale.edu/ModelDB/ | NeuronLand | C++ | 2 |
| HBP Morphology Viewer SWC +[13] | .swc | Neuromorpho.Org | Custom | Python | 1 |
| Imaris (Oxford Instruments, RRID:SCR_007370) | .ims | Neuromorpho.Org | NeuronLand (HDF5 Library) | C++ | 1 |
| KNOSSOS[8] | .nml[a] | Neuromorpho.Org | Custom | Python | 1 |
| Neuroglancer (RRID:SCR_015631) | .obj .stl .ply | http://fafb-ffn1.storage.googleapis.com/data.html | skeletor[59] | Python | 3 |
| Neurolucida[6, 20] | .asc | Neuromorpho.Org | NeuronLand;HBP[13 b] | C++; Node.js | 7 |
| Neurolucida | .dat | Neuromorpho.Org | NeuronLand; HBP[b] | C++; Node.js | 3 |
| Neurolucida | .nrx | Neuromorpho.Org | NeuronLand; HBP[b] | C++; Node.js | 1 |
| Neurolucida | .xml | Neuromorpho.Org | NeuronLand; HBP[b] | C++; Node.js | 1 |
| NeuroML[21, 22] | .nml[c] | Neuromorpho.Org | NeuronLand; Custom | C++; Python | 15 |
| NeuronJ[12] | .ndf | Neuromorpho.Org | Bonfire[31 b] | Octave | 4 |
| NEURON[15, 16] | .hoc | Neuromorpho.Org | Custom | Python | 11 |
| NeuroZoom[62] | .swc | Neuromorpho.Org | NeuronLand | C++ | 2 |
| NINDS3D[63] | .anat | Neuromorpho.Org | NeuronLand | C++ | 1 |
| PSICS[43] (RRID: SCR_014159) | .xml | psics.org/examples.html | NeuronLand | C++ | 1 |
| PyKNOSSOS[58] | .nmx | Neuromorpho.Org | Custom | Python | 1 |
| SNT TRACES[10, 11] | .traces | Neuromorpho.Org | FIJI[29] (SNT plugin[11]); Custom | Java; Python | 2 |
| TREES Toolbox[14] | .mtr | Neuromorpho.Org | TREES Toolbox[b] | Octave | 1 |
| TREES Toolbox | .mat | Neuromorpho.Org | TREES Toolbox[b] | Octave | 2 |
| Visualization Toolkit[64] | .vtk | natverse[d] | Custom | Python | 2 |

[a]KNOSSOS .nml format, while being an XML file, is not compliant with the NeuroML .nml format.
[b]Customized implementation.
[c]NeuroML recommends using .cell.nml for NeuroML v2 cell files[22], and .nml1 for NeuroML v1 files[21, 23].
[d]swc files were converted to .vtk for testing.

Conversion of volumetric mesh reconstructions to SWC is implemented using the "by_wavefront" method of the skeletor[59] Python library, which is capable of importing a wide variety of triangular mesh formats (https://trimsh.org/index.html). The imported mesh volume is first smoothed down using mesh contraction[60], then skeletonized, and finally post-processed to remove spurious nodes and branches. We have optimized the parameter values of the skeletonization algorithm to facilitate fast and accurate conversion for the majority of mesh reconstructions, but optionally users can also choose to set their own values by including a "mesh_config.txt" file when uploading the mesh files for conversion using the web interface. As an example for possible user modifications, we have also made the default configuration file publicly available (https://github.com/NeuroMorpho/xyz2swc/blob/main/xyz2swc/utils/mesh2swc_config.txt). The 'by_wavefront' algorithm calculates the SWC radius (R) by propagating a wave in the shape of a ring across the mesh surface, and subsequently collapsing all surface points on this ring into a single sample point located at the center of the ring. The plane of the ring is perpendicular to the direction of wave propagation. The radius of the SWC sample is the aggregate mean based on all points on the ring (with the option to change the mean into minimum, first quartile, median, third quartile, or maximum by modifying the configuration file).

The SWC standardization process is a two-part sequential operation consisting of (i) specification check, and (ii) error correction (Fig. 3). The specification checks and corrections (Table 3) are implemented using custom subroutines developed in Python. The subroutines iterate through the data fields of each sample point in the SWC to detect, and if possible then correct, values that do not meet standard specification. We implemented a sorting algorithm to target common non-standard formatting cases that include the first sample point not being a root; index numbering of the sample points not being sequential; or parent samples being referred to before they are defined. The sorting algorithm parses the SWC file in a looping sequence, alternating between rearranging the rows and re-numbering the Index and Parent fields of each row accordingly, while preserving the original tree structure, until standardization is achieved.

To detect a contour representation of the cell body, we use a curvature test (Fig. 5a). A soma section is a set of three or more contiguous soma sample points (identified by Type = 1), where the first point is a root (identified by Parent = −1) and the last point is either a bifurcation or a terminal. Specifically, let $\{A, P_1, P_2, \ldots, P_{n-2}, C\} \in \mathbb{R}^3$ denote the $n$ sample points along the soma section, such that $A$ is the first and $C$ is the last sample point. The 'remote' point B of the section is defined as the sample with the largest sum of distances from the first and last points,

$$B := \underset{i \le n-2}{\mathrm{argmax}} \left( ||A - P_i|| + ||C - P_i|| \right) \text{ for } n \ge 3, \quad (1)$$

where $|| \circ ||$ denotes the Euclidean distance between two points in $\mathbb{R}^3$. The curvature angle of the soma section is then defined as $\theta := \angle ABC$. The curvature test is repeated for all detected soma sections in the SWC file.

**Reporting summary**

Further information on research design is available in the Nature Portfolio Reporting Summary linked to this article.

## Data availability

All digital reconstruction data used in this study are publicly available for download from the NeuroMorpho.Org (version 8.4) repository at https://neuromorpho.org. The reconstruction files are made available both in the format that they were collected in as provided by the contributors, and in their equivalent SWC representations. In addition, for convenience, a small subset of the data sufficient to interpret and verify the functioning of the *xyz2swc* software is also deposited on GitHub: https://github.com/NeuroMorpho/xyz2swc/tree/main/input/to_convert.

## Code availability

The *xyz2swc* (RRID:SCR_023317) code[60] is open source and available at https://github.com/neuromorpho/xyz2swc. The latest stable version of the source code, libraries, modules, and all other dependencies needed to install and run the service locally is also published as a Docker image at https://hub.docker.com/r/neuromorpho/xyz2swc.

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

## Acknowledgements

This research was supported in part by NIH grants R01NS39600 (G.A.A.), R01NS86082 (G.A.A.), U01MH114829 (G.A.A.), and RF1MH128693 (G.A.A.). We thank all developers of the open-source software packages used in this project, notably Rembrandt Bakker (HBP Morphology Viewer), Hermann Cuntz (TREES Toolbox), Tiago Ferreira (SNT), Bonnie Firenstein (Bonfire), Greg Jefferis (natverse), Philipp Schlegel (Skeletor), and Adrian Wanner (pyKNOSSOS). We are indebted to our colleagues at the Center for Neural Informatics, Structures, and Plasticity (CN3) for their many insightful discussions. We are also deeply thankful to Dr. Alexander Bates for very constructively and collegially pointing the way to important improvements in this work, most notably the inclusion of synaptic connectivity and the challenging conversion from mesh form to SWC using existing open-source tools.

## Author contributions

K.M.: Conceptualization; Formal analysis; Investigation; Methodology; Software; Validation; Visualization; Writing—original draft; Writing—review & editing. Bengt Ljungquist: Conceptualization; Data curation; Investigation; Methodology; Software; Validation; Visualization; Writing—original draft; Writing—review & editing. James Ogden: Conceptualization; Methodology; Software; Validation; Writing—review & editing. S.N.: Conceptualization; Investigation; Software; Validation; Writing—original draft; Writing—review & editing. R.G.A.: Methodology; Software; Writing—review & editing. L.N.: Conceptualization; Investigation; Writing—review & editing. G.A.A.: Conceptualization; Formal analysis; Data curation; Funding acquisition; Investigation; Methodology; Project administration; Resources; Supervision; Validation; Visualization; Writing—original draft; Writing—review & editing.

## Competing interests

The authors declare no competing interests.
