## [Peer Review File · Nature Communications]

Online conversion of reconstructed neural morphologies into standardized SWC formatREVIEWER COMMENTS

Reviewer #1 (Remarks to the Author):

This article by Ketan et al provides a unique and important resource for researchers in the field of computational neuroscience. The specification is clear and well documented and the addition of software to facilitate conversions provides a helpful utility. I have tested a variety of files on the xyz2swc tool and can report accurate and intuitive conversions. Included in my testing was purposefully designed files containing errors with gaps in connections or aberrant numbering, and in each case was detected correctly. Obviously, with 23 different file formats and >68 format variations not all combinations were included in my testing but I think by having the code posted on Github will provide a useful platform for submitting and tracking any issues that users encounter. Finally, it might be worthwhile providing a short video explaining the specification and utility of the tools in an effort to bring in and orient new users into the field.

Reviewer #2 (Remarks to the Author):

This manuscript describes an initiative to formalize the specification of the widely used SWC format for neuronal morphology reconstructions, as well as presents a tool, xyz2swc, which can be used to convert multiple other formats into SWC, as well as convert existing SWC into a standardized, cleaned up format. The manuscript is well written, the results outlined well, and the xyz2swc tool described in sufficient detail. The code for the converter is open source and reasonably well documented. A number of suggested updates below would improve the manuscript.

While the core/original SWC format is the focus of this manuscript, there have been variations produced (eSWC, SWC+). While these are mentioned briefly here, it would be good to have more discussion on the current status of these and their relationship to this formalized SWC and the governing/standardization process behind it. Particularly as there is some overlap in the authors of this manuscript and those formats.

It would be great if at least one reference/URL could be given for each of the input formats supported. Some are mentioned (e.g. Arbor/Nevin, NeuroZoom) with no external reference to uniquely identify them.

Table 4 should ideally have in a separate column the actual file extension(s) used to identify each format. It does seem to have it for some in capitalized form in the first column. But not all. For example, how are Genesis files identified? Additionally, the suffix *.nml is used for most NeuroML files, though NML is specified for Knossos files, which are not NeuroML compliant. It may be best to use *.cell.nml for NeuroML v2 cell files: <https://docs.neuroml.org/Userdocs/Conventions#file-naming>, and *.nml1 for NeuroML version 1 files and update the converter as appropriate.

It would be good also to add a column to Table 4 showing which of these formats have been tested against data from NeuroMorpho.Org (i.e. Table 2). For the others which haven't been tested this way some indication of how the conversion has been tested for correctness should be given.

In the Methods, point B is defined as "the point along the soma section with median Index between the first and last points". This does not seem to be the case for point B in Fig 5a.

On line 298: bran -> brain.

It is mentioned that there is often a header section in SWC files of lines starting with # which contain metadata information. The same field names are frequently used across SWC files (e.g. ORIGINAL_SOURCE, CREATURE, CONTRIBUTOR). Would it not be useful to specify in this manuscript here a recommended set of fields which could be used for metadata across all SWC files?

Online access to the converter is good, but it is not always the best option for researchers/developers, they may prefer to do their conversions offline on their local machines. It is good that this option is there in the source code, through the use of Docker. However I didn't manage to get the Docker file to build an image successfully, as it requires Matlab runtime. The

"Local installation using Docker" step in the repository readme should ideally be updated with full details for this, as the manuscript claims that the framework can be added to or updated.

Also, it would be good to add some indication in the manuscript of the recommended/preferred/easiest way to add support for a new morphology format to the xyz2swc framework. Is extending Neuronland the recommended way, or Python for example?

Reviewer #3 (Remarks to the Author):

*** Review: Online conversion of reconstructed neural morphologies into standardised SWC format ***

Mehta and colleagues provide a new online tool to convert a range of described neuron morphology file formats (23) into SWC files (eponymous 'Stockley-Wheal-Cannon' files) (Cannon et al. 1998). SWC files are very much the 'CSV' of the neuroinformatics world. They encode 'skeleton' representations of neurons (a tree-graph representation where each vertex is a 3D point which are, when visualised, connected by lines). The authors' vision is that their tool will help researchers convert their data into this simple format from a range of sources, and in so doing provide a tool to help standardise the field.

Lightweight digital representations of neurons are very useful in a variety of analyses. For example, in order to compare the similarity of two neurons or split it into axon and dendrite using just the 'skeleton' of a neuron, rather than its full 3D structure as might be represented by a boundary mesh or voxel data, is sufficient and less computationally expensive (Costa et al. 2014; Schneider-Mizell et al. 2015).

The target users are neuroscientists who are collecting and/or analysing single neuron morphology data, and wish to convert it into SWC format before distributing the data or working with it computationally. This user might not be used to using a programming language in which they could re-write their data into SWC format, and so having a nifty online tool to help them do so is useful. Notably, there are many variants of the SWC format, and so having a tool that helps enforce a single view of SWC files could be helpful. A key way in which it would be helpful is that analysis software written for neuron morphology analysis could become less fragile to neuron input file types. I managed to test the site a little bit, and find that it works simply and largely effectively. I have also managed to write client code in R for its API (reviewer attached file, xyz2swc.R) after a bit of a struggle due to documentation issues (see below).

In sum, this work is a step towards better Findable, Accessible, Interoperable, and Reusable (FAIR) resources in neuroinformatics. The work first aims to (i) define a universal SWC file format and then (ii) presents a tool to help others convert into it. They achieve (ii) by using different 'modules' of code written by others that convert between file types, writing their own conversion scripts where they find gaps. This allows them to update individual modules as needed, it is an approach that might help them expand to cover new formats as they emerge.

There are a few issues that limit the impact of this work in my opinion:

*** Major Issues ***

Unclear why the given SWC format is better than others in use and how it is different from the original description

It would be helpful if the authors signposted where their description of their SWC file format differs from the original work (Cannon et al. 1998) and similar attempts such as SWCplus. If I naively search google for how to build a SWC file, the top hit is <https://neuroinformatics.nl/swcPlus/> followed by <http://www.neuronland.org/NLMorphologyConverter/MorphologyFormats/SWC/Spec.html>. The swcPlus file format as used by the HumanBrainProject is fundamentally similar to what the authors describe here. It is different in a few key ways. The authors seem to be inspired by it (they say their format is adapted from it <https://swc-specification.readthedocs.io/en/latest/swc.html>) but why not just use it? In addition, in their documentation the authors give different type field values

from the paper draft I see in front of me (<https://swc-specification.readthedocs.io/en/latest/swc.html> section 1). Here they say: "The basic set of types used in NeuroMorpho.org SWC files are: TypeID | Description — | — 0 | undefined 1 | soma 2 | axon 3 | (basal) dendrite 4 | apical dendrite 5 | custom 6 | unspecified neurite 7 | glia processes", assigning a label to 0 which is not assigned in the paper, and is converted to 6 by xyz2swc. There appears to be a governing board (with electoral features) that 'decides' the SWC format (for NeuroMorpho.org?) (<https://swc-specification.readthedocs.io/en/latest/governance.html>) but this is not mentioned or made clear in the paper. In sum, decisions that have gone into the 'standard' being presented are small in number but a little opaque and confused. In addition, the column names for the standardised SWC files given by xyz2swc (Index, TypeID, X, Y, Z, Radius, ParentIndex) are different from in the paper (Index, Type, X, Y, Z, R, Parent).

Broken links to key resources, code and documentation issues

The authors say: "We have made the SWC file format specification publicly available (<https://swc125specification.readthedocs.io>)". However, I get a 404 error. One the xyz2swc webpage, I am given the link <https://neuromorpho.org/xyz2swc/docs/> for documentation but I get: {"detail":"Not Found"} when I click on it. Also, the address <https://neuromorpho.org/xyz2swc/ui/> does not seem to be linked to from NeuroMorpho.org. Presumable it should live at HELP -> Tools & Links when the paper is published. The authors might want to consider making their 'Tools & Links' page on the main navigation bar of their website, rather than nested in HELP, so that users can find their tools more easily and so use them. The authors provide a Docker Container for their code in order to enable users to run it locally, the page would benefit from an overview on its use: <https://hub.docker.com/r/neuromorpho/xyz2swc>. The authors already have this on their nicely put together github page: <https://github.com/NeuroMorpho/xyz2swc>. They should, however, add citations for the tools they have used from others on their README (they do this in their paper itself well). When using using the tryout feature here (<https://neuromorpho.org/xyz2swc/docs#/convertfiles>), I get the error message "The requested resource is not available":

```
<html><head><title>Apache Tomcat/7.0.54 - Error report</title><style><!--H1 {font-family:Tahoma,Arial,sans-serif;color:white;background-color:#525D76;font-size:22px;} H2 {font-family:Tahoma,Arial,sans-serif;color:white;background-color:#525D76;font-size:16px;} H3 {font-family:Tahoma,Arial,sans-serif;color:white;background-color:#525D76;font-size:14px;} BODY {font-family:Tahoma,Arial,sans-serif;color:black;background-color:white;} B {font-family:Tahoma,Arial,sans-serif;color:white;background-color:#525D76;} P {font-family:Tahoma,Arial,sans-serif;background:white;color:black;font-size:12px;}A {color : black;}A.name {color : black;}HR {color : #525D76;}</style> </head><body><h1>HTTP Status 404 - /neuroMorpho/checkfiles</h1>
```

type Status report

message /neuroMorpho/checkfiles

description The requested resource is not available.

<h3>Apache Tomcat/7.0.54</h3></body></html>. I think the reason for this is that the API doc website is using the Request URL <https://neuromorpho.org/checkfiles>. However, if I use the Developer Tools to spy on what is happening when I use <https://neuromorpho.org/xyz2swc/ui> it appears to be using <https://neuromorpho.org/xyz2swc/checkfiles> NOT <https://neuromorpho.org/checkfiles>. I therefore tried to write code to engage the API to convert files for me from R. I have attached the code with this review, attempting to use the 'checkfiles' endpoint. However, using this end point in my R code yields an ERROR 500 "Internal Server Error". I managed to figure out that - contrary to the instruction given at <https://neuromorpho.org/xyz2swc/docs#/convertfiles>, a 'folder' field seems to be necessary. Once I made this change, the R code worked. I think the documentation needs to be improved in light of this. In addition, it would be helpful if the authors supply some code that uses in API (in python or R) in their README, or even as supplemental data for their paper, to show users how it can be done.

*** Enhancements to Seriously Consider ***

SWC as-is ignores synapses so might not be fit for purpose in the next decade

A key outcome of the paper is to present “a standardized specification of the SWC file format”. The format described is minimally different from other descriptions, which is suitable as the author’s aim is not to reinvent the wheel. However, I think there is a missed opportunity here to attempt to expand the utility of the format. For example, considering synapse data alongside morphology data. People are most interested in neurons precisely because they communicate with one another using synapses. Unlike mitochondria, cytoskeletal features or other cellular properties, synapses are a core feature commonly co-analysed with neuronal morphology, that are unique to neural cells. Because in the past it had been uncommon to capture spatial synapse information alongside single neuron reconstructions, e.g. from dye-fills, the SWC file format and its brethren do not consider synapses. However, we are now in an age of ever advancing connectomics. The NeuroMorpho database (Ascoli 2006; Ascoli, Donohue, and Halavi 2007) currently (Version 8.4.67) contains 245,626 neurons from a range of species, collected over decades. However, ~200,000 D. melanogaster neurons will soon (within months) be available from a single connectomic project (FAFB-FlyWire <https://ngl.flywire.ai/>) . ~24,000 are already available from the smaller ‘hemibrain’ project (<https://neuprint.janelia.org/>) . Whole brain D. melanogaster larval brain (Winding et al. 2023), *Platynereis dumerilii* (Randel et al. 2015), *C. elegans* (Cook et al. 2019) have added hundreds to thousands of neurons to the world supply of single-neuron morphologies. Hundreds of neurons are also yielded from extant vertebrate connectomic datasets, and we will soon see this number be thousands per data set, then tens of thousands and more. The majority of these reconstructions have manually annotated or (increasingly) automatically annotated synapses associated with them. Put simply, in the near future the majority of single neuron reconstructions will be accompanied by synaptic information (both pre- and post-, per neuron) from electron microscope or X-ray tomographic data sets. A general neuronal file format that does not appreciate this may struggle to find purchase if connectomic researchers opt to continue to use custom formats or a rival emergent standard, which does. The authors may tell me that including synaptic data is out of scope for the present work, and I accept that the work they have already done has utility without it. However, it really depends on the goal of the paper. Does the paper wish to only present an online conversion tool? Or does it wish to present an online conversion tool as a means of helping the field adopt a standardised file format that is fit for purpose over the next decade? I use the SWC file format for my morphological data, and .CSV files for the related synapse data - but this is a little unwieldy and each researcher is currently handling this differently. Mehta et al. could help nip this field format heterogeneity in the bud with very simple adjustments to their paper. Namely, envisioning a way to record synapse data in a SWC file (e.g. a table commented out at the end, so it can be read by some systems and ignored by others, idea from Philipp Schlegel) and demonstrating its use. I fear to do otherwise risks trying to implement a standard that will be ignored by those who want morphology+synapses in a single file. The ~24k neurons of the hemibrain project (Scheffer et al. 2020) could be taken as test case. (Of course, each project would still need to wrestle with other data annotations more specific to the given project and less ‘primary’ in the popular conceptualisation of what constitutes a neuron, e.g. mitochondria, gap junctions, cytoskeleton, vesicles, etc.).

The described standardised SWC file format does not consider/keep meta data on when converted through xyz2swc

Meta data could be include in the SWC file. In fact, in the original authors’ conceptualisation the file would start is a # and then then a standardised set of meta data entries (header). These would, for example, tell you what species and sample the data came from, the cell type captured and the spatial units for the 3D data. Including this would be extremely helpful in analysing neuronal data because the researcher would only need the data in the file itself without having to refer to documentation that described the data source. The authors may note to me that this meta data is missing for the majority of reconstructions and users of their tool may not wish to take the time to add it - but the fields can be left with NA or similar if the user does not want to fill them in. Indeed, many SWC files lack an entry for R (radius) as it can be difficult or unreliable to compute. Most critically the meta data should tell the user if points are in, say, micrometers or nanometers or some specific voxel units, and could also tell the user something about the reference frame. The

authors only briefly mention the possibility of meta data on line 112 and do not suggest a standard way of doing it. An issue they do not mention is that the meta data stored in the user's original file (possible in some of the file formats, i.e. as headers) will be lost. Having meta data as a part of the file standard would improve its credentials as a FAIR resource. Minimally any extant header or footers could be copied over into the 'standardised' SWC file.

Raw neuron morphology data sometimes exists only in mesh form and this conversion is a common challenge

Some neuron reconstruction environments, such a neuroglancer (now very popular) may only provide neuronal reconstructions for neurons as mesh volumes, for example as .OBJ files (but also .ply, .stl, .blend). Mesh or voxel representations can also be generated from software such a FIJI and Amira. Converting meshes into skeletons is actually the most common question I receive from scientists looking to convert their neural data into SWC. Many researchers could put together a script to convert one skeleton representation into another if they had to (again the convenience this new work adds is important and warranted) however converting a 3D mesh into a skeleton representation is much more involved. The authors could seriously consider improving their resource by adding a module to convert from volume to skeleton. An example of an effective (esp. 'wave' method) and well kept repository that does just that, can be found here: <https://pypi.org/project/skeletor/> (Zenodo: <https://zenodo.org/record/5138552#.ZD1X9ezMIeY>). Since the authors current work uses modules built by others for some of their file conversions, it may be simple for them to include skeletonisation from meshes by incorporating this or similar as a module - though some parameters for skeletonisation would need to be chosen on the website's front-end. I think this is a reasonable and in-scope request that would improve the impact of the paper.

*** Medium Questions and Corrections ***

The authors say: "Multiple SWC variants have emerged, causing confusion among neuroscientists and slowing down research progress." I think this claim about slowing down research progress is a little overblown. One of the things people have liked about SWC files is that they can add their own columns to keep track of custom information. I am not sure how one would evidence this claim.

The 'Type' column is an integer value that maps onto a compartment type for the point being described. In the literature, Types 0-4 is quite consistent. The authors note that '5' can be custom, though in reality any number greater than 7 in their schema could have a custom assignment? And this gives the user far more flexibility. What is the utility xyz2swc setting all numbers greater than 7 to 6? Reserving only 5 for custom works only if I have just a single extra custom label I want. What if I want to add 'cell body fibre tract' and 'synaptic twig' for example, I could have used 8 and 9, but xyz2swc would flag those as issues. If there is a historic reason for this, can it please be cited or explained?

Why would 'glia processes' be 7? Sounds like quite a custom, study dependent field, seems odd to specify glial processes when by far the most common use case is for SWC files to describe a neuron. If there is a historic reason for this, can it please be cited or explained?

Type = 0 is commonly used to mean undefined. The authors prefer 6, and in their standardisation they convert 0 to 6. E.g. here: <https://neuroinformatics.nl/swcPlus/> and here: <http://www.neuronland.org/NLMorphologyConverter/MorphologyFormats/SWC/Spec.html>. Because these are top hits on Google for the file format I suspect many user, myself included, use 0. It feels more fitting. What is the reasoning for using 6? If there is a historic reason for this, can it please be cited or explained? Confusingly, the authors' own documentation seem to indicate that 0 can be 'undefined' cable: <https://swc-specification.readthedocs.io/en/latest/swc.html>. A table for 'Type' values would be easier to digest.

The SWC file description should define how R (radius) is best calculated. Is it, for example, the smallest distance between a surface point and the point in the SWC file? Or does one determine the direction of travel for the neuron at that point, and draw a line perpendicular to it, or many lines and take an average of the radius? Or what? Having a suggestion for calculating R would be helpful.

Some, but not all, software that deal with SWC files require the file start with the root node (Parent: -1). The authors standard seems to abide by this, but it is not explicitly mentioned in the text for section: Standardized Specification for the SWC File Format. It is mentioned/implied later in the text.

I think one common-enough file format that has been missed in .vtk. This file format can be used for neuron visualisation in paraview (<https://www.paraview.org/>) and is the file format needed to use the warping object based registration software deformetrica (<https://www.deformetrica.org/>) (Durrleman et al. 2014). I have attached some .vtk file samples for the authors with this review. The R package nat is capable for this conversion (nat::read.neurons).

xyz2swc is said to automatically insert default values when it is missing something for the SWC file format. The authors give an example, "the NeuronJ NDF format does not store the thickness and depth of each branch, capturing the arbor as a two-dimensional linear projection. The resultant SWC file is given a uniform Radius of 0.5 and Z coordinates of 0". I worry that a naive user - or more likely a secondary user working with that person's converted file - might think filled in values are real. I can see why Z would need to be 0 in this case (the inputted file format is 2D) but setting the radius to a seemingly arbitrary value (0.5) rather than NA is issuesome. I know the reason the authors wish to do this is because, for example, 3D plotting software might expect a R value and will not render it if given a NA. However, that is an issue for the plotting software and must be fixed on that end. Injecting false values is dangerous. Another option would be adding a footer to the file to note inserted values. The github documentation and the xyz2swc web-page should give all instances in which a default value is inject and what that value is, and perhaps why (say, table 3 in the paper).

What are the data storage trade-offs? Could the authors compare the file sizes of neuron data before and after conversion (where meta data / other data is not dropped)? A key consideration as reconstructions become more prevalent might be storage.

I ran some tests of my own. I tried to give xyz2swc the 303 .xml field that define the C. elegans connectome (available here:<https://github.com/openworm/CElegansNeuroML/tree/master/CElegans/generatedMorphML>). It can convert all of them. However, sometimes the website freezes and does not complete the conversion of ~1-2 neurons, they get stuck on 'waiting'. I think the issue might be with large files. Some SWC files for highly detailed EM reconstructions can reach into the tens of megabytes. Can a more helpful message be given to users? Perhaps a timeout if the file takes too long? It seems to either work quickly or not at all, for the same file. The website seems to be overwhelmed quite quickly when someone tries to use it in parallel (i.e. multiple users at once). If I open two browsers and try to convert a single file in both it gets stuck on 'Waiting' for a long time (possibly indefinitely) even with only a 10 kB file (April 17th at 17:20 EST).

*** Minor Thoughts ***

Provide a definition for "ASCII encoded" for the casual reader (line 111).

From a 'branding' perspective the authors might benefit from a short hand way to referring to their standardised SWC format as apart from the ecosystem of slightly different SWC file formats that exist. For example, 'sSWC' for 'standardised SWC'? In a similar manner to how HBP has SWCplus.

Could figure 1 be improved to contain - in one space - all the information need to fundamentally define a SWC file? I.e. also include the Type to label table and a short description of the columns. This would give a single image that can be shared between scientists to help the viewer immediately get to grips with the format (like an infographic).

All figures seem to be at a slightly low resolution?

Would it be possible to run xyz2swc in reverse, and convert to other skeleton file formats? This would be useful if the code you wanted to run did not take SWC files, e.g. Paraview or FIJI. This would increase the utility of the resource.

*** Conclusion ***

This paper assists in moving the field towards an even more expansive adoption of SWC as a FAIR way of working with neuronal morphology data. Mehta et al. (i) define the standard SWC file and (ii) provide an online web tool for converting into this standard, and checking SWC files that might not be in standard form.

Concerning (i) - While useful as a clarification this is not an advance over what has been described/been in use before, both in the work of NeuroMorpho and other groups, such as HBP with SWCplus. Why different decisions have been made versus, say, SWCplus is not clear to me. I feel that the authors have an opportunity here to - rather than redefine SWCs very slightly - instead, do that AND provide a ground plan for how SWC can adapt for the future. Specifically, how they might be modified to handle meta data and synapses in a flexible but standard way. With the advent of connectomics, a single file that can capture morphology+synapses+meta would be very useful. I would suggest capturing meta data in the header, and synapses in the footer after a flag. SWC have been popular because they are easily humanly interpretable but also because they are flexible. Users can add new columns and new values into the Type/Label column in order to better describe their neurons of study. I think that the authors have aimed to prevent this because their tool wipes out added columns, headers and footers and does not allow Type entries above 7. Concerning (ii) - the most difficult challenge for people wanting to analyse neuronal morphology is going from a mesh to a SWC. Having this feature I think is the difference between an underused web tool and a popular one. Converting between different skeleton file types is still very useful of course. At the moment conversions are all one way (into standardised SWC). Bi-directional conversions would also be useful. Also, I think there might be an issue with getting the API to work correctly (I could have made a mistake though) as mentioned above.

I recommend that this paper be published (!!!) after minor corrections. I understand that the three 'enhancement' points I have made (namely guidance for including synapses and meta data and a module for volume-to-skeleton conversions) involve an expansion of scope the authors might rather resist. But I feel this is warranted (e.g. synaptic reconstructions will outnumber non-synaptic ones by next year) and also reasonably straightforward to accomplish. Am happy to discuss further with the authors. Without any of those three it is hard to make a case for progress from the status quo. The other issues I have raised in my 'main' and 'medium' sections above I think do need to be addressed.

My best wishes,

Alexander Bates

References

- Ascoli, Giorgio A. 2006. "Mobilizing the Base of Neuroscience Data: The Case of Neuronal Morphologies." *Nature Reviews. Neuroscience* 7 (4): 318-24.
- Ascoli, Giorgio A., Duncan E. Donohue, and Maryam Halavi. 2007. "NeuroMorpho.Org: A Central Resource for Neuronal Morphologies." *The Journal of Neuroscience: The Official Journal of the Society for Neuroscience* 27 (35): 9247-51.
- Cannon, R. C., D. A. Turner, G. K. Pyapali, and H. V. Wheal. 1998. "An on-Line Archive of Reconstructed Hippocampal Neurons." *Journal of Neuroscience Methods* 84 (1-2): 49-54.
- Cook, Steven J., Travis A. Jarrell, Christopher A. Brittin, Yi Wang, Adam E. Bloniarz, Maksim A. Yakovlev, Ken C. Q. Nguyen, et al. 2019. "Whole-Animal Connectomes of Both Caenorhabditis Elegans Sexes." *Nature* 571 (7763): 63-71.
- Costa, Marta, Aaron D. Ostrovsky, James D. Manton, Steffen Prohaska, and Gregory S. X. E. Jefferis. 2014. "NBLAST: Rapid, Sensitive Comparison of Neuronal Structure and Construction of Neuron Family Databases." 2014.
- Durrleman, Stanley, Marcel Prastawa, Nicolas Charon, Julie R. Korenberg, Sarang Joshi, Guido Gerig, and Alain Trounev. 2014. "Morphometry of Anatomical Shape Complexes with Dense Deformations and Sparse Parameters." *NeuroImage* 101 (November): 35-49.
- Randel, Nadine, Réza Shahidi, Csaba Verasztó, Luis A. Bezares-Calderón, Steffen Schmidt, and Gáspár Jékely. 2015. "Inter-Individual Stereotypy of the Platynereis Larval Visual Connectome."

eLife 4 (June): e08069.

Scheffer, Louis K., C. Shan Xu, Michal Januszewski, Zhiyuan Lu, Shin-Ya Takemura, Kenneth J. Hayworth, Gary B. Huang, et al. 2020. "A Connectome and Analysis of the Adult Drosophila Central Brain." eLife 9 (September). <https://doi.org/10.7554/eLife.57443>.

Schneider-Mizell, Casey M., Stephan Gerhard, Mark Longair, Tom Kazimiers, Feng Li, Maarten F. Zwart, Andrew Champion, et al. 2015. "Quantitative Neuroanatomy for Connectomics in Drosophila." bioRxiv, 026617.

Winding, Michael, Benjamin D. Pedigo, Christopher L. Barnes, Heather G. Patsolic, Youngser Park, Tom Kazimiers, Akira Fushiki, et al. 2023. "The Connectome of an Insect Brain." Science 379 (6636): eadd9330.

Reviewer #4 (Remarks to the Author):

This paper describes a pragmatic and practical approach to standardisation of data for a specific, but large community of researchers. It addresses a clear problem in the domain and demonstrates its effectiveness by converting all existing data and evaluating the results.

The application is niche and only applicable for researchers involved in neural morphology work. However, good FAIR data solutions need to address specific concerns in a particular domain. Adhering to the FAIR principles will enable these results, or this approach, to be applicable for the broader scientific community.

The paper would benefit from a discussion of related work. It is a novel approach for this field, but there are a number of similar approaches in other fields. For example, the Open Microscopy Environment adopts a similar file format conversion approach for managing differences between microscope vendors (Goldberg, I., C. Allan, J.-M. Burel, D. Creager, A. Falconi, H. Hochheiser, J. Johnston, J. Mellen, P.K. Sorger, and J.R. Swedlow. (2005) The Open Microscopy Environment (OME) Data Model and XML File: Open Tools for Informatics and Quantitative Analysis in Biological Imaging. Genome Biol. 6:R47). There are also a number of different file format conversion approaches in omics integration and also systems biology tools, conversion to SBML.

The strength of the approach is its simplicity (as mentioned by the authors). It is a syntactic conversion of heterogeneous formats to a canonical common representation, which is therefore easier to compare and reuse.

The authors state that they adhere to the FAIR principles, but it would strengthen the paper to describe specifically which aspects of FAIR they address and which need improvement. This approach is clearly a step forward, but does it address all the semantic requirements of FAIR?

The main weakness of this approach is that it focuses on syntactic representation without addressing the semantics of what is being represented. Metadata and semantics are at the heart of the FAIR principles. The standardised SWC format has space in the header to add metadata, but the format and minimum requirements for this are not specified. It may not be possible to automatically extract and populate standardised metadata from the legacy database data, but it would be advantageous to suggest a common set of metadata terms and a format. Perhaps this should be done in collaboration with NeuroMorpho.Org, but setting out how this would be done in the future and how this would be useful would strengthen the paper.

The link to the documentation is broken – (<https://swc-125specification.readthedocs.io/>), but the link from the application works (<https://swc-specification.readthedocs.io/en/latest/>). Please change this.

Response to Reviewers' Comments

Online conversion of reconstructed neural morphologies into standardized SWC format

Mehta, Ketan, Ljungquist, Bengt, Ogden, James, Nanda, Sumit, Ascoli, Ruben G., Ng, Lydia, Ascoli, Giorgio A.

We thank the editor and the reviewers for their useful suggestions to improve the content and presentation of our research, and the time they have spent with our manuscript. We have revised the paper addressing every comment as summarized below **in bold fonts**. All changes to the manuscript are in blue text for easier visibility.

Reviewer #1

R1.1 This article by Ketan et al provides a unique and important resource for researchers in the field of computational neuroscience. The specification is clear and well documented and the addition of software to facilitate conversions provides a helpful utility. I have tested a variety of files on the xyz2swc tool and can report accurate and intuitive conversions. Included in my testing was purposefully designed files containing errors with gaps in connections or aberrant numbering, and in each case was detected correctly. Obviously, with 23 different file formats and >68 format variations not all combinations were included in my testing but I think by having the code posted on Github will provide a useful platform for submitting and tracking any issues that users encounter. Finally, it might be worthwhile providing a short video explaining the specification and utility of the tools in an effort to bring in and orient new users into the field.

We are very grateful for the positive comments, for taking the time to test the tool with a variety of files, and for creating a custom data set designed to detect format inconsistencies. We are glad the testing confirmed the accuracy of the conversion, and we are pleased you found the user experience intuitive. Thanks for the helpful recommendation to provide a short explanatory video. We have created two 1-minute videos, which are now linked to the xyz2swc main page (<https://neuromorpho.org/xyz2swc/ui/>): one overviewing the SWC format specification, and the other demonstrating how to use the online xyz2swc converter. The videos are also linked to the GitHub page and the specification documentation page.

Reviewer #2

R2.1 This manuscript describes an initiative to formalize the specification of the widely used SWC format for neuronal morphology reconstructions, as well as presents a tool, xyz2swc, which can be used to convert multiple other formats into SWC, as well as convert existing SWC into a standardized, cleaned up format. The manuscript is well written, the results outlined well, and the xyz2swc tool described in sufficient detail. The code for the converter is open source and reasonably well documented. A number of suggested updates below would improve the manuscript.

Thank you for the careful review, the positive comments, and the constructive suggestions, which we followed to further improve the manuscript as detailed below.

R2.2 While the core/original SWC format is the focus of this manuscript, there have been variations produced (eSWC, SWC+). While these are mentioned briefly here, it would be good to have more discussion on the current status of these and their relationship to this formalized SWC and the governing/standardization process behind it. Particularly as there is some overlap in the authors of this manuscript and those formats.

Good point. We have expanded the Conclusions (lines 375-383) to provide additional information on the status of eSWC and SWC+ as well as their relationship to this formalized SWC standard.

R2.3 It would be great if at least one reference/URL could be given for each of the input formats supported. Some are mentioned (e.g. Arbor/Nevin, NeuroZoom) with no external reference to uniquely identify them.

We agree and have updated both the text and Table 4 to provide at least one reference and/or RRID for each file format.

*R2.4 Table 4 should ideally have in a separate column the actual file extension(s) used to identify each format. It does seem to have it for some in capitalized form in the first column. But not all. For example, how are Genesis files identified? Additionally, the suffix *.nml is used for most NeuroML files, though NML is specified for Knossos files, which are not NeuroML compliant. It may be best to use *.cell.nml for NeuroML v2 cell files: <https://docs.neuroml.org/Userdocs/Conventions#file-naming>, and *.nml1 for NeuroML version 1 files and update the converter as appropriate.*

Thanks for the excellent suggestion. File extensions have not always been kept consistent across independent labs, but we have now modified and completed Table 4 with an additional column making this information explicit case-by-case. We have no control on the extension used by Knossos but have added a footnote to Table 4 with the recommendation regarding various .nml variants.

R2.5 It would be good also to add a column to Table 4 showing which of these formats have been tested against data from NeuroMorpho.Org (i.e. Table 2). For the others which haven't been tested this way some indication of how the conversion has been tested for correctness should be given.

We have modified Table 4 as recommended.

R2.6 In the Methods, point B is defined as “the point along the soma section with median Index between the first and last points”. This does not seem to be the case for point B in Fig 5a.

We appreciate your catching this inconsistency. Figure 5a is indeed correct, and we have now revised the text (lines 467-472) with the right description.

R2.7 On line 298: bran -> brain.

Thanks! Corrected.

R2.8 It is mentioned that there is often a header section in SWC files of lines starting with # which contain metadata information. The same field names are frequently used across SWC files (e.g. ORIGINAL_SOURCE, CREATURE, CONTRIBUTOR). Would it not be useful to specify in this manuscript here a recommended set of fields which could be used for metadata across all SWC files?

Yes, it would be. We hesitated in this regard when submitting the original manuscript because the community has not yet reached a consensus on minimum metadata requirements. We have now added a section of the paper describing “recommendations” to be considered optional at this time, as opposed to “requirements” for standard compliance. The metadata header is the first such optional recommendation. We will bring up these recommendations for discussion at the first SWC standard governing board meeting after the paper is published for possible inclusion in v1.1.0 of the requirements.

R2.9 Online access to the converter is good, but it is not always the best option for researchers/developers, they may prefer to do their conversions offline on their local machines. It is good that this option is there in the source code, through the use of Docker. However I didn't manage to get the Docker file to build an image successfully, as it requires Matlab runtime. The “Local installation using Docker” step in the repository readme should ideally be updated with full details for this, as the manuscript claims that the framework can be added to or updated.

Thank you for raising this issue. We have updated the ReadMe file (<https://github.com/NeuroMorpho/xyz2swc>) and now provide full details for the local installation. To further simplify the installation and use of xyz2swc on local machines, we have also revised the code replacing all Matlab modules with Octave, thus eliminating the need for Matlab runtime libraries.

R2.10 Also, it would be good to add some indication in the manuscript of the recommended/preferred/easiest way to add support for a new morphology format to the xyz2swc framework. Is extending Neuronland the recommended way, or Python for example?

Great idea. Based on our experience, contributing Python modules is the most effective way to add support, and have indicated this in the revised manuscript (lines 425-432).

Reviewer #3

*R3.1 *** Review: Online conversion of reconstructed neural morphologies into standardised SWC format ***
Mehta and colleagues provide a new online tool to convert a range of described neuron morphology file formats (23) into SWC files (eponymous ‘Stockley-Wheal-Cannon’ files) (Cannon et al. 1998). SWC files are very much the ‘CSV’ of the neuroinformatics world. They encode ‘skeleton’ representations of neurons (a tree-graph representation where each vertex is a 3D point which are, when visualised, connected by lines). The authors’ vision is that their tool will help researchers convert their data into this simple format from a range of sources, and in so doing provide a tool to help standardise the field. Lightweight digital representations of neurons are very useful in a variety of analyses. For example, in order to compare the similarity of two neurons or split it into axon and dendrie using just the ‘skeleton’ of a neuron, rather than its full 3D structure as might be represented by a boundary mesh or voxel data, is sufficient and less computationally expensive (Costa et al. 2014; Schneider-Mizell et al. 2015). The target users are neuroscientists who are collecting and/or analysing single neuron morphology data, and wish to convert it into SWC format before distributing*

the data or working with it computationally. This user might not be used to using a programming language in which they could re-write their data into SWC format, and so having a nifty online tool to help them do so is useful. Notably, there are many variants of the SWC format, and so having a tool that helps enforce a single view of SWC files could be helpful. A key way in which it would be helpful is that analysis software written for neuron morphology analysis could become less fragile to neuron input file types. I managed to test the site a little bit, and find that it works simply and largely effectively. I have also managed to write client code in R for its API (reviewer attached file, xyz2swc.R) after a bit of a struggle due to documentation issues (see below). In sum, this work is a step towards better Findable, Accessible, Interoperable, and Reusable (FAIR) resources in neuroinformatics. The work first aims to (i) define a universal SWC file format and then (ii) presents a tool to help others convert into it. They achieve (ii) by using different ‘modules’ of code written by others that convert between file types, writing their own conversion scripts where they find gaps. This allows them to update individual modules as needed, it is an approach that might help them expand to cover new formats as they emerge.

Thank you very much for the careful and positive review, the hands-on testing, and the constructive suggestions, which we followed to further improve the manuscript and the tool as detailed below.

*R3.2 There are a few issues that limit the impact of this work in my opinion: *** Major Issues *** Unclear why the given SWC format is better than others in use and how it is different from the original description - It would be helpful if the authors signposted where their description of their SWC file format differs from the original work (Cannon et al. 1998) and similar attempts such as SWCplus.*

Thank you for the comment. We have edited the Conclusions to further clarify the motivation for (“why”) and the distinction of (“how”) the presented SWC standard. In short, we tried to follow the original Cannon et al 1998 work in the overall simplicity, but we had to make choices when multiple SWC variants emerged in part due to the lack of a complete specification in the Cannon et al 1998 description – this is why a formal standard is needed! The specific choices were based largely on the frequency of usage, as determined by the number of data files in any of the SWC variants received by NeuroMorpho.Org.

R3.3 If I naively search google for how to build a SWC file, the top hit is <https://neuroinformatics.nl/swcPlus/> followed by <http://www.neuronland.org/NLMorphologyConverter/MorphologyFormats/SWC/Spec.html>. The swcPlus file format as used by the HumanBrainProject is fundamentally similar to what the authors describe here.

We believe that Google searches are not so much naïve but potentially misleading in their result ranking, because the “top hits” vary not only depending on the exact wording of the query, but also critically on the user search history, geographical location, the time of the query, and several other factors. This makes an objective discussion based on the order of Google results difficult if not impossible.

R3.4 It is different in a few key ways. The authors seem to be inspired by it (they say their format is adapted from it <https://swc-specification.readthedocs.io/en/latest/swc.html>) but why not just use it?

We mentioned SWC+ in the specification because we follow the same style of documentation and we wanted to provide due credit. The inspiration, however, goes the other way around: the SWC+ description (URL provided by referee) is itself adapted from NeuronLand (whose author is also a

coauthor of this manuscript), and NeuronLand has always used NeuroMorpho.Org as the testing reference for its format. Historically, among those 3 resources, NeuroMorpho.Org came first (2006-ongoing) followed by NeuronLand (2008-2016) and SWC+ (2016-2019). We chose not to use SWC+ as the standard because, based on the usage frequency, this format does not seem to have gained substantial traction in the community so far. We also gathered informal feedback both from NeuroMorpho.Org users and during a dedicated discussion session in the BICCN anatomy working group, which revealed that many experimental anatomists find the XML formalism adopted by SWC+ less intuitively accessible. We now briefly explain in the Conclusion why we chose not to use SWC+ as the standard.

R3.5 In addition, in their documentation the authors give different type field values from the paper draft I see in front of me (<https://swc-specification.readthedocs.io/en/latest/swc.html> section 1). Here they say: “The basic set of types used in NeuroMorpho.org SWC files are: TypeID | Description — | — 0 | undefined 1 | soma 2 | axon 3 | (basal) dendrite 4 | apical dendrite 5 | custom 6 | unspecified neurite 7 | glia processes”, assigning a label to 0 which is not assigned in the paper, and is converted to 6 by xyz2swc.

We are grateful for the pointers to possible sources of confusion. We now clarify in the paper (lines 103-111) the distinction between type 0 (undetermined) and type 6 (unspecified neurite). In particular, type 0 is used when the structural domain is left undetermined in the reconstruction. An SWC point with type 0 could correspond to any structural domain, including soma or glia process. Type 6 is used when the structural domain is certainly a neurite (dendrites or axon), but the available information is insufficient to discriminate between the exact type of neurite. This is often the case in dissociated cultures, in early developmental stages (when axons and dendrites are not fully differentiated), and in some invertebrate neural systems where the same neurite can occasionally serve as both dendrite and axon. We have also corrected the xyz2swc code so that it now outputs Type 0 as default instead of type 6 when the structural domain is completely unknown.

R3.6 There appears to be a governing board (with electoral features) that ‘decides’ the SWC format (for NeuroMorpho.org?) (<https://swc-specification.readthedocs.io/en/latest/governance.html>) but this is not mentioned or made clear in the paper. In sum, decisions that have gone into the ‘standard’ being presented are small in number but a little opaque and confused.

The revised manuscript now mentions the governing board and explains its functions (lines 159-164).

R3.7 In addition, the column names for the standardised SWC files given by xyz2swc (Index, TypeID, X, Y, Z, Radius, ParentIndex) are different from in the paper (Index, Type, X, Y, Z, R, Parent).

We have corrected the xyz2swc code so as to make the column names consistent with those indicated in the paper.

R3.8 Broken links to key resources, code and documentation issues - The authors say: “We have made the SWC file format specification publicly available (<https://swc125specification.readthedocs.io/>)”. However, I get a 404 error.

This appears to be an issue of the Nature Communication manuscript handling system, which automatically adds line numbers when generating the pdf. The ‘125’ string in the URL is spurious, but the URL (<https://swc-specification.readthedocs.io>) is correct in the submitted manuscript and we will ensure it is correctly type-set at the proof stage.

R3.9 On the [xyz2swc](https://neuromorpho.org/xyz2swc) webpage, I am given the link <https://neuromorpho.org/xyz2swc/docs/> for documentation but I get: {"detail": "Not Found"} when I click on it.

Thanks for noticing this issue, which we have now corrected (note that the correct link is <https://neuromorpho.org/xyz2swc/docs> without / at the end).

R3.10 Also, the address <https://neuromorpho.org/xyz2swc/ui/> does not seem to be linked to from [NeuroMorpho.org](https://neuromorpho.org). Presumably it should live at HELP -> Tools & Links when the paper is published. The authors might want to consider making their ‘Tools & Links’ page on the main navigation bar of their website, rather than nested in HELP, so that users can find their tools more easily and so use them.

Thanks, we have added the link as recommended and we plan to move the parent “Tools & Links” page to the main navigation bar at the next major release upon publication of the paper.

R3.11 The authors provide a Docker Container for their code in order to enable users to run it locally, the page would benefit from an overview on its use: <https://hub.docker.com/r/neuromorpho/xyz2swc>. The authors already have this on their nicely put together github page: <https://github.com/NeuroMorpho/xyz2swc>. They should, however, add citations for the tools they have used from others on their README (they do this in their paper itself well).

Great suggestions. We have added the usage overview in [hub.docker](https://hub.docker.com/r/neuromorpho/xyz2swc) and added citations for the used tools in the GitHub ReadMe file.

R3.12 When using using the tryout feature here (<https://neuromorpho.org/xyz2swc/docs#/convertfiles>), I get the error message

```
“The requested resource is not available”: <html><head><title>Apache Tomcat/7.0.54 - Error report</title><style><!--H1 {font-family:Tahoma,Arial,sans-serif;color:white;background-color:#525D76;font-size:22px;} H2 {font-family:Tahoma,Arial,sans-serif;color:white;background-color:#525D76;font-size:16px;} H3 {font-family:Tahoma,Arial,sans-serif;color:white;background-color:#525D76;font-size:14px;} BODY {font-family:Tahoma,Arial,sans-serif;color:black;background-color:white;} B {font-family:Tahoma,Arial,sans-serif;color:white;background-color:#525D76;} P {font-family:Tahoma,Arial,sans-serif;background:white;color:black;font-size:12px;}A {color : black;}A.name {color : black;}HR {color : #525D76;}</style> </head>
```

HTTP Status 404 - /neuroMorpho/checkfiles

*type*Status report

message[/neuroMorpho/checkfiles](https://neuromorpho.org/xyz2swc/docs#/convertfiles)

*description*The requested resource is not available.

Apache Tomcat/7.0.54

```
</body></html>.
```

I think the reason for this is that the API doc website is using the Request URL <https://neuromorpho.org/checkfiles>. However, if I use the Developer Tools to spy on what is happening when I use <https://neuromorpho.org/xyz2swc/docs#/convertfiles>.

/neuromorpho.org/xyz2swc/ui it appears to be using <https://neuromorpho.org/xyz2swc/checkfiles> NOT <https://neuromorpho.org/checkfiles>. I therefore tried to write code to engage the API to convert files for me from R. I have attached the code with this review, attempting to use the 'checkfiles' endpoint. However, using this end point in my R code yields an ERROR 500 "Internal Server Error". I managed to figure out that - contrary to the instruction given at <https://neuromorpho.org/xyz2swc/docs#/convertfiles>, a 'folder' field seems to be necessary. Once I made this change, the R code worked. I think the documentation needs to be improved in light of this. In addition, it would be helpful if the authors supply some code that uses in API (in python or R) in their README, or even as supplemental data for their paper, to show users how it can be done.

Thank you very much for the thorough testing and feedback. We corrected the API doc website, improved the documentation as suggested, and modified the API for greater robustness. Moreover, we now provide Python code to use the API in the ReadMe file, as recommended, and uploaded the example to GitHub.

*R3.13 *** Enhancements to Seriously Consider ****

*SWC as-is ignores synapses so might not be fit for purpose in the next decade - A key outcome of the paper is to present "a standardized specification of the SWC file format". The format described is minimally different from other descriptions, which is suitable as the author's aim is not to reinvent the wheel. However, I think there is a missed opportunity here to attempt to expand the utility of the format. For example, considering synapse data alongside morphology data. People are most interested in neurons precisely because they communicate with one another using synapses. Unlike mitochondria, cytoskeletal features or other cellular properties, synapses are a core feature commonly co-analysed with neuronal morphology, that are unique to neural cells. Because in the past it had been uncommon to capture spatial synapse information alongside single neuron reconstructions, e.g. from dye-fills, the SWC file format and its brethren do not consider synapses. However, we are now in an age of ever advancing connectomics. The NeuroMorpho database (Ascoli 2006; Ascoli, Donohue, and Halavi 2007) currently (Version 8.4.67) contains 245,626 neurons from a range of species, collected over decades. However, ~200,000 *D. melanogaster* neurons will soon (within months) be available from a single connectomic project (FAFB-FlyWire <https://ngl.flywire.ai/>). ~24,000 are already available from the smaller 'hemibrain' project (<https://neuprint.janelia.org/>). Whole brain *D. melanogaster* larval brain (Winding et al. 2023), *Platynereis dumerilii* (Randel et al. 2015), *C. elegans* (Cook et al. 2019) have added hundreds to thousands of neurons to the world supply of single-neuron morphologies. Hundreds of neurons are also yielded from extant vertebrate connectomic datasets, and we will soon see this number be thousands per data set, then tens of thousands and more. The majority of these reconstructions have manually annotated or (increasingly) automatically annotated synapses associated with them. Put simply, in the near future the majority of single neuron reconstructions will be accompanied by synaptic information (both pre- and post-, per neuron) from electron microscope or X-ray tomographic data sets. A general neuronal file format that does not appreciate this may struggle to find purchase if connectomic researchers opt to continue to use custom formats or a rival emergent standard, which does. The authors may tell me that including synaptic data is out of scope for the present work, and I accept that the work they have already done has utility without it. However, it really depends on the goal of the paper. Does the paper wish to only present an online conversion tool? Or does it wish to present an online conversion tool as a means of helping the field adopt a standardised file format that is fit for purpose over the next decade? I use the SWC file format for my morphological data, and .CSV files for the related synapse data - but this is a little unwieldy and each researcher is currently handling this differently. Mehta et al. could help nip this field format heterogeneity in the bud with very simple adjustments to their paper. Namely, envisioning a way to record synapse data in a SWC file (e.g. a table commented out at the end, so it can be read by some systems*

and ignored by others, idea from Philipp Schlegel) and demonstrating its use. I fear to do otherwise risks trying to implement a standard that will be ignored by those who want morphology+synapses in a single file. The ~24k neurons of the hemibrain project (Scheffer et al. 2020) could be taken as test case. (Of course, each project would still need to wrestle with other data annotations more specific to the given project and less 'primary' in the popular conceptualisation of what constitutes a neuron, e.g. mitochondria, gap junctions, cytoskeleton, vesicles, etc.).

We appreciate the in-depth commentary. We had in fact proposed a similar method as Philipp Schlegel's idea (namely, additional information commented out at the end of the SWC file, so it can be read by some systems and ignored by others) in our earlier technique to account for time-lapse data and molecular distributions (doi.org/10.1038/sdata.2017.207). We have already used precisely that kind of representation for our own datasets: see e.g. header (line 3) and footer (starting at line 6662) at neuromorpho.org/dableFiles/ascoli/Source-Version/form3-OE_L5D6_MT_Fact.eswc

We did not initially include this possibility in the paper because as of now most SWC data are from light microscopy, and we could not be sure of what the EM community would consider a reasonably effective representation for synapses. However, based on your feedback, we are now encouraged that it may be sensible to try just that at this time. We greatly appreciated your willingness "to discuss further with the authors", and are deeply grateful for the examples CSV files of synaptic data associated with corresponding SWC files of morphologies you kindly shared with us. As you recommended, we now propose to append the relevant data content to the SWC files with appropriate explanations in the footer comments, as outlined in the new "Recommendations for Optional Inclusion of Ancillary Information in SWC Files" section of the Results (line 165). We also include selected examples as supplemental materials, cite the references you indicated (thanks), and of course acknowledge your help and valuable prompts in the paper.

R3.14 The described standardised SWC file format does not consider/keep meta data on when converted through xyz2swc - Meta data could be include in the SWC file. In fact, in the original authors' conceptualisation the file would start is a # and then then a standardised set of meta data entries (header). These would, for example, tell you what species and sample the data came from, the cell type captured and the spatial units for the 3D data. Including this would be extremely helpful in analysing neuronal data because the researcher would only need the data in the file itself without having to refer to documentation that described the data source. The authors may note to me that this meta data is missing for the majority of reconstructions and users of their tool may not wish to take the time to add it - but the fields can be left with NA or similar if the user does not want to fill them in. Indeed, many SWC files lack an entry for R (radius) as it can be difficult or unreliable to compute. Most critically the meta data should tell the user if points are in, say, micrometers or nanometers or some specific voxel units, and could also tell the user something about the reference frame. The authors only briefly mention the possibility of meta data on line 112 and do not suggest a standard way of doing it. An issue they do not mention is that the meta data stored in the user's original file (possible in some of the file formats, i.e. as headers) will be lost. Having meta data as a part of the file standard would improve its credentials as a FAIR resource. Minimally any extant header or footers could be copied over into the 'standardised' SWC file.

Although the community has not yet reached a consensus on minimum metadata requirements, we agree with this sentiment. We have now added a section of the paper describing "recommendations" to be considered "optional" at this time, as opposed to "requirements" for standard compliance. The metadata header is the first such optional recommendation. We will

bring up these recommendations for discussion at the first SWC standard governing board meeting after the paper is published for possible inclusion in v1.1.0 of the requirements. Moreover, we have modified xyz2swc so as to copy over (retain) the header/footer content into the standardized SWC files, as recommended.

R3.15 Raw neuron morphology data sometimes exists only in mesh form and this conversion is a common challenge - Some neuron reconstruction environments, such a neuroglancer (now very popular) may only provide neuronal reconstructions for neurons as mesh volumes, for example as .OBJ files (but also .ply, .stl, .blend). Mesh or voxel representations can also be generated from software such a FIJI and Amira. Converting meshes into skeletons is actually the most common question I receive from scientists looking to convert their neural data into SWC. Many researchers could put together a script to convert one skeleton representation into another if they had to (again the convenience this new work adds is important and warranted) however converting a 3D mesh into a skeleton representation is much more involved. The authors could seriously consider improving their resource by adding a module to convert from volume to skeleton. An example of an effective (esp. 'wave' method) and well kept repository that does just that, can be found here: <https://pypi.org/project/skeletor/> (Zenodo: <https://zenodo.org/record/5138552#.ZD1X9ezMIeY>). Since the authors current work uses modules built by others for some of their file conversions, it may be simple for them to include skeletonisation from meshes by incorporating this or similar as a module - though some parameters for skeletonisation would need to be chosen on the website's front-end. I think this is a reasonable and in-scope request that would improve the impact of the paper.

Thank you for the suggestion and the pointer. We have added a new module to xyz2swc that allows conversion of mesh files to .swc. The module uses the 'by_wavefront' method of <https://pypi.org/project/skeletor/> to skeletonize triangle mesh volumes, along with performing some pre- and post-processing. With the addition of this new module, xyz2swc now supports the popular .obj, .stl, .ply formats used by Neuroglancer and other similar environments (and in theory can convert all mesh formats supported by <https://trimsh.org/index.html>). We have optimized the skeletonization parameter values for optimal conversion, but optionally users can also choose to set their own values by including a 'mesh_config.txt' when uploading the mesh files for conversion using the web interface. As an example for possible user modifications, we have also made the default configuration file publicly available (https://github.com/NeuroMorpho/xyz2swc/blob/main/xyz2swc/utils/mesh2swc_config.txt). We have updated the manuscript text (lines 434-443), Fig. 3, and Table 4 to reflect these changes.

*R3.16 *** Medium Questions and Corrections ****

The authors say: "Multiple SWC variants have emerged, causing confusion among neuroscientists and slowing down research progress." I think this claim about slowing down research progress is a little overblown. One of the things people have liked about SWC files is that they can add their own columns to keep track of custom information. I am not sure how one would evidence this claim.

Customizing information is indeed useful for independent labs analyzing their own data. However, it may become problematic when sharing data through mass database distribution. Moreover, the proliferation of variants does slow down development of open-source tools meant to maximize interoperability. We have modified the manuscript (line 61-65) to qualify our assertion and to mention the experience of the computational neuroscience user community and related software ecosystem.

R3.17 The 'Type' column is an integer value that maps onto a compartment type for the point being described. In the literature, Types 0-4 is quite consistent. The authors note that '5' can be custom, though in reality any number greater than 7 in their schema could have a custom assignment? And this gives the user far more flexibility. What is the utility xyz2swc setting all numbers greater than 7 to 6? Reserving only 5 for custom works only if I have just a single extra custom label I want. What if I want to add 'cell body fibre tract' and 'synaptic twig' for example, I could have used 8 and 9, but xyz2swc would flag those as issues. If there is a historic reason for this, can it please be cited or explained?

This is a compelling argument. After careful consideration, we have decided to modify the code so as to not flag or convert types higher than 7. We have also edited the manuscript (Table 1) and the online documentation accordingly.

R3.18 Why would 'glia processes' be 7? Sounds like quite a custom, study dependent field, seems odd to specify glial processes when by far the most common use case is for SWC files to describe a neuron. If there is a historic reason for this, can it please be cited or explained?

Although for many years use of SWC files was practically limited to describing neurons, application to glia began no later than in 2017 and grew dramatically in more recent times. Currently, >27% of NeuroMorpho.Org content is glia, the largest contributing lab (Sandra Siegert's) is a glia lab, and the most represented cell type in the entire database is microglia. We now explain this data surge in the revised manuscript (lines 96-103) to support the requirement for a standardized type dedicated to glia processes.

R3.19 Type = 0 is commonly used to mean undefined. The authors prefer 6, and in their standardisation they convert 0 to 6. E.g. here: <https://neuroinformatics.nl/swcPlus/> and here: <http://www.neuronland.org/NLMorphologyConverter/MorphologyFormats/SWC/Spec.html>. Because these are top hits on Google for the file format I suspect many user, myself included, use 0. It feels more fitting. What is the reasoning for using 6? If there is a historic reason for this, can it please be cited or explained? Confusingly, the authors' own documentation seem to indicate that 0 can be 'undefined' cable: <https://swc-specification.readthedocs.io/en/latest/swc.html>.

We are grateful for the pointers to possible sources of confusion. We now clarify in the paper (lines 103-111) the distinction between type 0 (undefined) and type 6 (unspecified neurite). Specifically, type 0 is used when the structural domain is left undetermined in the reconstruction. An SWC point with type 0 could correspond to any structural domain, including soma or glia process. Type 6 is used when the structural domain is certainly a neurite (dendrites or axon), but the available information is insufficient to discriminate between the exact type of neurite. This is often the case in dissociated cultures, in early developmental stages (when axons and dendrites are not fully differentiated), and in some invertebrate nervous systems where the same neurite can occasionally serve as both dendrite and axon. We have also corrected the xyz2swc code so that it now outputs Type 0 as default instead of type 6 when the structural domain is completely unknown.

R3.20 A table for 'Type' values would be easier to digest.

We agree, and have included the recommended additional Table (Table 1). We have also modified the online documentation to display the Type values as in the revised manuscript.

R3.21 The SWC file description should define how R (radius) is best calculated. Is it, for example, the smallest distance between a surface point and the point in the SWC file? Or does one determine the direction of travel for the neuron at that point, and draw a line perpendicular to it, or many lines and take an average of the radius? Or what? Having a suggestion for calculating R would be helpful.

Thank you for the useful suggestion. Not all skeletonization algorithms output a radius for the SWC samples. The “by_wavfront” algorithm outputs a radius by using a propagating wave (in the shape of a ring) across the mesh surface, and subsequently collapsing all surface points on this ring into a single SWC sample point located at the center of the ring. The plane of the ring is perpendicular to the direction of wave propagation. The radius of the SWC sample is the aggregate mean based on all points on the ring (with the option to change the mean into min, 1st quartile, median, 3rd quartile, or max by modifying the configuration file. We have added this content to the Materials and Methods section of the revised manuscript (lines 443-449), to be discussed by the governing board for future inclusion in the standard SWC requirements.

R3.22 Some, but not all, software that deal with SWC files require the file start with the root node (Parent: - 1). The authors standard seems to abide by this, but it is not explicitly mentioned in the text for section: Standardized Specification for the SWC File Format. It is mentioned/implied later in the text.

Thank you for pointing out this lacuna. The requirement is now explicitly mentioned in the revised manuscript (lines 118-119).

R3.23 I think one common-enough file format that has been missed in .vtk. This file format can be used for neuron visualisation in paraview (<https://www.paraview.org/>) and is the file format needed to use the warping object based registration software deformetrica (<https://www.deformetrica.org/>) (Durrleman et al. 2014). I have attached some .vtk file samples for the authors with this review. The R package nat is capable for this conversion (nat::read.neurons).

Thank you for spotting the missing format, and for providing suitable examples for testing. We have modified xyz2swc to allow conversion of .vtk data to .swc (and appropriately updated Fig. 3 and Table 4). Note, however, that we were unable to read VTK files using the natverse package. The natverse documentation also seems to indicate that the package can only export as .vtk, but not import .vtk files (cf. table on bottom of page at <https://natverse.org/nat/reference/fileformats.html>). Instead, we have developed a custom Python script to directly read the VTK polydata file structure (<https://pypi.org/project/vtk/>) and convert it into SWC.

R3.24 xyz2swc is said to automatically insert default values when it is missing something for the SWC file format. The authors give an example, “the NeuronJ NDF format does not store the thickness and depth of each branch, capturing the arbor as a two-dimensional linear projection. The resultant SWC file is given a uniform Radius of 0.5 and Z coordinates of 0”. I worry that a naive user - or more likely a secondary user working with that person’s converted file - might think filled in values are real. I can see why Z would need to be 0 in this case (the inputted file format is 2D) but setting the radius to a seemingly arbitrary value (0.5) rather than NA is issuesome. I know the reason the authors wish to do this is because, for example, 3D

plotting software might expect a R value and will not render it if given a NA. However, that is an issue for the plotting software and must be fixed on that end. Injecting false values is dangerous. Another option would be adding a footer to the file to note inserted values. The github documentation and the xyz2swc web-page should give all instances in which a default value is inject and what that value is, and perhaps why (say, table 3 in the paper).

We fully appreciate this concern. As the referees realizes, this design maximizes interoperability with popular community software. We have thus opted to accept the suggestion of adding a footer to the file to note inserted values, and we have updated Table 3, the GitHub ReadMe page, and the xyz2swc code accordingly, as recommended.

R3.25 What are the data storage trade-offs? Could the authors compare the file sizes of neuron data before and after conversion (where meta data / other data is not dropped)? A key consideration as reconstructions become more prevalent might be storage.

Thank you for raising an interesting and relevant point. We have added an analysis of file size (lines 296-305) and updated Table 2 based on available NeuroMorpho.Org content in the revised manuscript.

R3.26 I ran some tests of my own. I tried to give xyz2swc the 303 .xml field that define the C. elegans connectome (available here: <https://github.com/openworm/CElegansNeuroML/tree/master/CElegans/generatedMorphML>). It can convert all of them. However, sometimes the website freezes and does not complete the conversion of ~1-2 neurons, they get stuck on 'waiting'. I think the issue might be with large files. Some SWC files for highly detailed EM reconstructions can reach into the tens of megabytes. Can a more helpful message be given to users? Perhaps a timeout if the file takes too long? It seems to either work quickly or not at all, for the same file. The website seems to be overwhelmed quite quickly when someone tries to use it in parallel (i.e. multiple users at once). If I open two browsers and try to convert a single file in both it gets stuck on 'Waiting' for a long time (possibly indefinitely) even with only a 10 kB file (April 17th at 17:20 EST).

Thank you for the testing report. We have optimized the service backend to allow parallel access. Moreover, we have improved the message display to indicate expected conversion duration and timeout errors.

*R3.27 *** Minor Thoughts ****

Provide a definition for "ASCII encoded" for the casual reader (line 111).

Thanks – we have edited the manuscript as suggested (lines 123-126).

R3.28 From a 'branding' perspective the authors might benefit from a short hand way to referring to their standardised SWC format as apart from the ecosystem of slightly different SWC file formats that exist. For example, 'sSWC' for 'standardised SWC'? In a similar manner to how HBP has SWCplus.

We appreciate the suggestion. We have discussed this matter among coauthors as well as colleagues from the BICCN anatomy working group. The consensus was to not introduce a short-hand, because the risk to create confusion was deemed greater than the potential benefit. However, now that the first version of the manuscript has undergone peer review, we have formalized a versioning system for the SWC standard. Thus, we have started referring to the

current description as SWC v1.0.0, which helps set the standard apart from other SWC variations. Future deliberations of the governing board will produce versions v1.1.0 and beyond.

R3.29 Could figure 1 be improved to contain - in one space - all the information need to fundamentally define a SWC file? I.e. also include the Type to label table and a short description of the columns. This would give a single image that can be shared between scientists to help the viewer immediately get to grips with the format (like an infographic).

Great idea. We have improved Figure 1 as proposed.

R3.30 All figures seem to be at a slightly low resolution?

This appears to be an issue with the pdf conversion of the journal submission system, but the original figures should be available to the referees. At any rate, we will ensure that the figures appear in high resolution in the proofs.

R3.31 Would it be possible to run xyz2swc in reverse, and convert to other skeleton file formats? This would be useful if the code you wanted to run did not take SWC files, e.g. Paraview or FIJI. This would increase the utility of the resource.

We appreciate the suggestion. At this point, it is not possible to run xyz2swc in reverse. However, that functionality is already provided by the NeuronLand converter for the formats supported by that module. As part of the xyz2swc launch, we are also releasing the source code of NeuronLand, which should increase the utility of the resource. The revised manuscript now mentions this important element in the Materials and Methods (lines 429-432).

R3.32 *** Conclusion ***

This paper assists in moving the field towards an even more expansive adoption of SWC as a FAIR way of working with neuronal morphology data. Mehta et al. (i) define the standard SWC file and (ii) provide an online web tool for converting into this standard, and checking SWC files that might not be in standard form. Concerning (i) - While useful as a clarification this is not an advance over what has been described/been in use before, both in the work of NeuroMorpho and other groups, such as HBP with SWCplus. Why different decisions have been made versus, say, SWCplus is not clear to me. I feel that the authors have an opportunity here to - rather than redefine SWCs very slightly - instead, do that AND provide a ground plan for how SWC can adapt for the future. Specifically, how they might be modified to handle meta data and synapses in a flexible but standard way. With the advent of connectomics, a single file that can capture morphology+synapses+meta would be very useful. I would suggest capturing meta data in the header, and synapses in the footer after a flag. SWC have been popular because they are easily humanly interpretable but also because they are flexible. Users can add new columns and new values into the Type/Label column in order to better describe their neurons of study. I think that the authors have aimed to prevent this because their tool wipes out added columns, headers and footers and does not allow Type entries above 7.

Thanks again for the very thorough review. As described above, we have heeded your constructive advice and suggest a ground plan for how SWC can adapt for the future through modifications to capture in a single file metadata (in the header) and synapses (in the footer) in

addition to morphology in the main body. We have also modified the xyz2swc code to preserve extra information and to allow type entries above 7.

R3.33 Concerning (ii) - the most difficult challenge for people wanting to analyse neuronal morphology is going from a mesh to a SWC. Having this feature I think is the difference between an underused web tool and a popular one. Converting between different skeleton file types is still very useful of course. At the moment conversions are all one way (into standardised SWC). Bi-directional conversions would also be useful. Also, I think there might be an issue with getting the API to work correctly (I could have made a mistake though) as mentioned above.

We have made our best efforts to incorporate the conversion from mesh to SWC following the specific recommendation provided. We have also improved the API documentation and double-checked its correct functioning, with gratitude for the invaluable feedback.

R3.34 I recommend that this paper be published (!!!) after minor corrections. I understand that the three 'enhancement' points I have made (namely guidance for including synapses and meta data and a module for volume-to-skeleton conversions) involve an expansion of scope the authors might rather resist. But I feel this is warranted (e.g. synaptic reconstructions will outnumber non-synaptic ones by next year) and also reasonably straightforward to accomplish. Am happy to discuss further with the authors. Without any of those three it is hard to make a case for progress from the status quo. The other issues I have raised in my 'main' and 'medium' sections above I think do need to be addressed.

My best wishes, Alexander Bates (alexander_bates@hms.harvard.edu)

To the best of our understanding, we have addressed all criticisms - thank you again.

*R3.35 *** References ****

Ascoli, Giorgio A. 2006. "Mobilizing the Base of Neuroscience Data: The Case of Neuronal Morphologies." Nature Reviews. Neuroscience 7 (4): 318–24.

Ascoli, Giorgio A., Duncan E. Donohue, and Maryam Halavi. 2007. "NeuroMorpho.Org: A Central Resource for Neuronal Morphologies." The Journal of Neuroscience: The Official Journal of the Society for Neuroscience 27 (35): 9247–51.

Cannon, R. C., D. A. Turner, G. K. Pyapali, and H. V. Wheal. 1998. "An on-Line Archive of Reconstructed Hippocampal Neurons." Journal of Neuroscience Methods 84 (1-2): 49–54.

Cook, Steven J., Travis A. Jarrell, Christopher A. Brittin, Yi Wang, Adam E. Bloniarz, Maksim A. Yakovlev, Ken C. Q. Nguyen, et al. 2019. "Whole-Animal Connectomes of Both Caenorhabditis Elegans Sexes." Nature 571 (7763): 63–71.

Costa, Marta, Aaron D. Ostrovsky, James D. Manton, Steffen Prohaska, and Gregory S. X. E. Jefferis. 2014. "NBLAST: Rapid, Sensitive Comparison of Neuronal Structure and Construction of Neuron Family Databases." 2014.

Durrleman, Stanley, Marcel Prastawa, Nicolas Charon, Julie R. Korenberg, Sarang Joshi, Guido Gerig, and Alain Trounev. 2014. "Morphometry of Anatomical Shape Complexes with Dense Deformations and Sparse Parameters." NeuroImage 101 (November): 35–49.

Randel, Nadine, Réza Shahidi, Csaba Verasztó, Luis A. Bezares-Calderón, Steffen Schmidt, and Gáspár Jékely. 2015. "Inter-Individual Stereotypy of the Platynereis Larval Visual Connectome." eLife 4 (June): e08069.

Scheffer, Louis K., C. Shan Xu, Michal Januszewski, Zhiyuan Lu, Shin-Ya Takemura, Kenneth J. Hayworth,

Gary B. Huang, et al. 2020. "A Connectome and Analysis of the Adult *Drosophila* Central Brain." *eLife* 9 (September). <https://doi.org/10.7554/eLife.57443>.

Schneider-Mizell, Casey M., Stephan Gerhard, Mark Longair, Tom Kazimiers, Feng Li, Maarten F. Zwart, Andrew Champion, et al. 2015. "Quantitative Neuroanatomy for Connectomics in *Drosophila*." *bioRxiv*, 026617.

Winding, Michael, Benjamin D. Pedigo, Christopher L. Barnes, Heather G. Patsolic, Youngser Park, Tom Kazimiers, Akira Fushiki, et al. 2023. "The Connectome of an Insect Brain." *Science* 379 (6636): eadd9330.

We now cite relevant references to support the substantial additions in the revised manuscript.

Reviewer #4

R4.1 This paper describes a pragmatic and practical approach to standardisation of data for a specific, but large community of researchers. It addresses a clear problem in the domain and demonstrates its effectiveness by converting all existing data and evaluating the results. The application is niche and only applicable for researchers involved in neural morphology work. However, good FAIR data solutions need to address specific concerns in a particular domain. Adhering to the FAIR principles will enable these results, or this approach, to be applicable for the broader scientific community.

Thank you for the expert reading and positive remarks.

R4.2 The paper would benefit from a discussion of related work. It is a novel approach for this field, but there are a number of similar approaches in other fields. For example, the Open Microscopy Environment adopts a similar file format conversion approach for managing differences between microscope vendors (Goldberg, I., C. Allan, J.-M. Burel, D. Creager, A. Falconi, H. Hochheiser, J. Johnston, J. Mellen, P.K. Sorger, and J.R. Swedlow. (2005) *The Open Microscopy Environment (OME) Data Model and XML File: Open Tools for Informatics and Quantitative Analysis in Biological Imaging*. *Genome Biol.* 6:R47). There are also a number of different file format conversion approaches in omics integration and also systems biology tools, conversion to SBML.

We appreciate the suggestion and have expanded the discussion of related work in the Conclusions of the revised manuscript, including the recommended citation.

R4.3 The strength of the approach is its simplicity (as mentioned by the authors). It is a syntactic conversion of heterogeneous formats to a canonical common representation, which is therefore easier to compare and reuse. The authors state that they adhere to the FAIR principles, but it would strengthen the paper to describe specifically which aspects of FAIR they address and which need improvement. This approach is clearly a step forward, but does it address all the semantic requirements of FAIR?

Thanks for the recommendation. We have expanded the Results to address this point (lines 144-155).

R4.4 The main weakness of this approach is that it focuses on syntactic representation without addressing the semantics of what is being represented. Metadata and semantics are at the heart of the FAIR principles. The standardised SWC format has space in the header to add metadata, but the format and minimum requirements for this are not specified. It may not be possible to automatically extract and populate

standardised metadata from the legacy database data, but it would be advantageous to suggest a common set of metadata terms and a format. Perhaps this should be done in collaboration with NeuroMorpho.Org, but setting out how this would be done in the future and how this would be useful would strengthen the paper.

Although the community has not yet reached a consensus on minimum metadata requirements, we agree with this sentiment. We have now added a section of the paper describing “recommendations” to be considered “optional” at this time, as opposed to “requirements” for standard compliance. The metadata header is the first such optional recommendation. We will bring up these recommendations for discussion at the first SWC standard governing board meeting after the paper is published for possible inclusion in v1.1.0 of the requirements.

R4.5 The link to the documentation is broken – (<https://swc-125specification.readthedocs.io/>), but the link from the application works (<https://swc-specification.readthedocs.io/en/latest/>). Please change this.

This appears to be an issue of the Nature Communication manuscript handling system, which automatically adds line numbers when generating the pdf. The ‘125’ string in the URL is spurious, but the URL (<https://swc-specification.readthedocs.io/>) is correct in the submitted manuscript and we will ensure it is correctly type-set at the proof stage.

REVIEWERS' COMMENTS

Reviewer #2 (Remarks to the Author):

This version addresses all of my issues and plenty of others the other reviewers have raised, making a much improved manuscript.

Reviewer #3 (Remarks to the Author):

Mehta and colleagues have responded very well to the first round of reviews and amended their work such that I strongly recommend publication in Nature Communications.

I think I threw up a few fairly major suggestions in my last review, and a large number of medium things that should be addressed, and the authors have tackled them all directly and capably. Having provided my name, the authors also contacted me for ideas, sample data and clarifications by email. I thank them for kindly adding me to their acknowledgements. I think the paper looks great.

I note a few other major positive changes on the back of comments from other reviewers: I think the inclusion of a video outlining the SWC standardised file type and their methods, as recommended by reviewer 1, is a helpful addition to the work. Especially because it appears when the 'usage demo' button is clicked on <https://neuromorpho.org/xyz2swc/ui/>. The updates to Table 4 as recommended by reviewer 2 have made this paper a better reference for the position of the subfield and guide to the described resource. The author's replacement of MATLAB modules with open-source Octave ones at a suggestion from reviewer 2 better enables this work as an open-source tool in its own right.

The authors have also satisfactorily answered all of my points, including clarifying how their format differs from other versions of SWC (I did not know the history they provided in their rebuttal), the distinction between Type 0 and 6 fields, the governing board (though, minorly, I would add it's not clear who elects 2 of its members from the paper which sort of feels like a relevant methods thing), the wealth of glial data that rationalises glial processes being one of the standard SWC labels. They have also provided a definition of the root node, included a table of type values (Table 1), a SWC definition infographic (Figure 1), fixed issues related to URLs and visibility and added citations to their GitHub README. I found their added column on file size relative to SWC in Table 2 interesting and illuminating. I also liked the author's idea of versioning their standard definition, i.e. this paper describes SWC v1.0.0.

They have changed their tools in answer to my points: to default to Type 0 when the identity of a neuron point is unknown, to add inserted default values to the footer, to carry over header (e.g. metadata) and footer (e.g. synapses) information after standardisation and to not curtail custom field usage by allowing the Label column to exceed the value 6, improved the backend to support parallelisation and added more detailed error messages.

They have also taken very seriously my suggestion to try to skeletonise meshes by using Philipp Schlegel's python tool 'skeletor' as a module (they nicely cite the library with its zenodo DOI), sensibly choosing the fast wavefront (waves=1) method as a default but allowing the user to add more options with a config file. I think this addition to their tool has a high potential to help a number of researchers who want to skeletonise their data but are not experienced enough in R/python/similar to deploy these tools quickly by themselves.

However, this feature is not mentioned prominently on their Github page (<https://github.com/NeuroMorpho/xyz2swc/tree/main>), which also does not mention the mesh_config.txt file. Mesh file type conversion capability is listed in the file formats table that they have, but I think mentioning it in the first paragraph of the repo is helpful (they could have a slightly more expansive tldr). It also is not clear how the mesh_config.txt file is included when uploading a .obj file - does it just have to be one of the files selected? This is implied in the paper itself but should be clarified on the tool website and the Github repo so that users easily understand this without reading much.

I tried to convert two mesh .obj files from the flywire project. These files were ~200Mb and the conversion would take a long time but to reassure the user, a message that states mesh conversions can 'take up to tens of minutes to complete' would be useful, as most will think the tool is stuck. In my case, I did not know. I saw the yellow 'waiting' button (no progress bar) for many minutes. Eventually, a red

bar appeared but the yellow waiting button remained. Perhaps the files were too large or there was some other error? Could any python error suffered be printed to the website? I left it for an hour, but it did not complete. I tested with skeletor locally and it could skeletonise them. One suggestion if the file size is too large, would be to simplify the mesh before passing it to skeletor so that the author's implementation can handle it. Another minor point here is that the .obj has to have the mesh's normals encoded for the wavefront algorithm to work (I think). Many .obj files do not include this data as a mesh can be rendered without it. I tested also with .obj files with their normals stripped. Skeletor throws a warning when it cannot find the normals, but the xyz2swc web-interface does not give any indication as to the issue. The naive user would just assume the tool does not work, and move on. I have attached to this review sample .obj data that I tried and failed to skeletonise with the xyz2swc UI - raw meshes for AOTU19 neurons from the flywire and hemibrain drosophila connectome projects, and simplified versions.

I did a bit of debugging and wrote some R functions for API access to xyzswc in my last review and made it available here: <https://github.com/natverse/neuromorphr/blob/master/R/xyz2swc.R>. The authors made corrections to their API and have written their own python script for the same I think, they say in their rebuttal: "we now provide Python code to use the API in the ReadMe file, as recommended, and uploaded the example to GitHub". However, I cannot find it on their README? I see only a link to the API docs (<https://neuromorpho.org/xyz2swc/docs>). API access will help power users make use of their tool, providing an example means people do not have to figure out its use by themselves.

After the initial suggestion on synapses, and then a discussion and sample data exchange with the authors by email, the authors now give a brief description of how synapse information can be added to the footer, with specific suggestions on columns (one small update though is to add references 39 and 40 to the description of F2 in line 513, and whatever citations F1 needs on 512). I think this is a great addition, but would ask the authors to make the minor adjustment of including a supplementary table/figure, similar to their current Table or Figure 1, so as to visually show the suggestion. These visual aids are both helpful and influential in determining what conventions potential users - who may only skim-read the manuscript - eventually adopt. I think doing so would make the suggestion stronger. Ideally, this would also be done for the metadata suggestion (they could be combined into a single display item to save space). Not least because it was a quibble for 3 out of 4 reviewers. I would also recommend including these images on the README for the Github repo and relevant Neuromorpho.org page. This will help the suggested conventions be considered.

In summary, the paper should be published. I think the authors might want to consider my suggestions for their error messages/figures/tables/README to help maximise the number of adopters in the next years.

Best wishes and well done!,

Alex Bates

Reviewer #4 (Remarks to the Author):

This paper describes a pragmatic and practical approach to standardisation of data for a specific, but large community of researchers. It addresses a clear problem in the domain and demonstrates its effectiveness by converting all existing data and evaluating the results. The application is only applicable for researchers involved in neural morphology work, but this seems to be an active community who would benefit from such standardisation. The authors aim is to adhere to the FAIR principles and much more clearly state how they do that - and also how this work relates to other initiatives in related fields. What is learned from the use of this approach will be applicable for the broader scientific community developing FAIR data solutions.

The authors have addressed my previous comments and concerns (and also many of the concerns of the other reviewers). The manuscript is clearer and has been improved as a result

Response to Reviewers' Comments

Online conversion of reconstructed neural morphologies into standardized SWC format

Mehta, Ketan, Ljungquist, Bengt, Ogden, James, Nanda, Sumit, Ascoli, Ruben G., Ng, Lydia, Ascoli, Giorgio A.

We thank the editor and the reviewers for their useful suggestions to improve the content and presentation of our research, and the time they have spent with our manuscript. We have revised the paper addressing every comment as summarized below **in bold fonts**. All changes to the manuscript are in **green text** for easier visibility.

Reviewer #2

R2.1 This version addresses all of my issues and plenty of others the other reviewers have raised, making a much improved manuscript.

Thank you for the positive comment.

Reviewer #3

R3.1 Mehta and colleagues have responded very well to the first round of reviews and amended their work such that I strongly recommend publication in Nature Communications.

I think I threw up a few fairly major suggestions in my last review, and a large number of medium things that should be addressed, and the authors have tackled them all directly and capably. Having provided my name, the authors also contacted me for ideas, sample data and clarifications by email. I thank them for kindly adding me to their acknowledgements. I think the paper looks great.

I note a few other major positive changes on the back of comments from other reviewers: I think the inclusion of a video outlining the SWC standardised file type and their methods, as recommended by reviewer 1, is a helpful addition to the work. Especially because it appears when the 'usage demo' button is clicked on <https://neuromorpho.org/xyz2swc/ui/>. The updates to Table 4 as recommended by reviewer 2 have made this paper a better reference for the position of the subfield and guide to the described resource. The author's replacement of MATLAB modules with open-source Octave ones at a suggestion from reviewer 2 better enables this work as an open-source tool in its own right.

The authors have also satisfactorily answered all of my points, including clarifying how their format differs from other versions of SWC (I did not know the history they provided in their rebuttal), the distinction between Type 0 and 6 fields, the governing board (though, minorly, I would add it's not clear who elects 2 of its members from the paper which sort of feels like a relevant methods thing), the wealth of glial data that rationalises glial processes being one of the standard SWC labels. They have also provided a definition of the root node, included a table of type values (Table 1), a SWC definition infographic (Figure 1), fixed issues

related to URLs and visibility and added citations to their GitHub README. I found their added column on file size relative to SWC in Table 2 interesting and illuminating. I also liked the author's idea of versioning their standard definition, i.e. this paper describes SWC v1.0.0.

They have changed their tools in answer to my points: to default to Type 0 when the identity of a neuron point is unknown, to add inserted default values to the footer, to carry over header (e.g. metadata) and footer (e.g. synapses) information after standardisation and to not curtail custom field usage by allowing the Label column to exceed the value 6, improved the backend to support parallelisation and added more detailed error messages.

They have also taken very seriously my suggestion to try to skeletonise meshes by using Philipp Schlegel's python tool 'skeletor' as a module (they nicely cite the library with its zenodo DOI), sensibly choosing the fast wavefront (waves=1) method as a default but allowing the user to add more options with a config file. I think this addition to their tool has a high potential to help a number of researchers who want to skeletonise their data but are not experienced enough in R/python/similar to deploy these tools quickly by themselves.

Thank you very much for the careful and positive review, the hands-on testing, and the constructive suggestions which allowed us to further improve the manuscript.

R3.2 They have also taken very seriously my suggestion to try to skeletonise meshes by using Philipp Schlegel's python tool 'skeletor' as a module (they nicely cite the library with its zenodo DOI), sensibly choosing the fast wavefront (waves=1) method as a default but allowing the user to add more options with a config file. I think this addition to their tool has a high potential to help a number of researchers who want to skeletonise their data but are not experienced enough in R/python/similar to deploy these tools quickly by themselves. However, this feature is not mentioned prominently on their Github page (<https://github.com/NeuroMorpho/xyz2swc/tree/main>), which also does not mention the mesh_config.txt file. Mesh file type conversion capability is listed in the file formats table that they have, but I think mentioning it in the first paragraph of the repo is helpful (they could have a slightly more expansive tldr). It also is not clear how the mesh_config.txt file is included when uploading a .obj file - does it just have to be one of the files selected? This is implied in the paper itself but should be clarified on the tool website and the Github repo so that users easily understand this without reading much.

We agree and have revised the GitHub ReadMe file to now include a clearly visible list of Features (<https://github.com/neuromorpho/xyz2swc/#features-tldr>) in the very first section of the file (which also serves as a tldr). The ability of xyz2swc to skeletonize mesh files is now mentioned in this newly added list of Features. We have also clarified and elaborated on the use of mesh_config.txt by including it within the Supported Tools and Formats section of the ReadMe (<https://github.com/neuromorpho/xyz2swc/#supported-tools-and-formats>).

R3.3 I tried to convert two mesh .obj files from the flywire project. These files were ~200Mb and the conversion would take a long time but to reassure the user, a message that states mesh conversions can 'take up to tens of minutes to complete' would be useful, as most will think the tool is stuck. In my case, I did not know. I saw the yellow 'waiting' button (no progress bar) for many minutes. Eventually, a red bar appeared but the yellow waiting button remained. Perhaps the files were too large or there was some other error? Could any python error suffered be printed to the website? I left it for an hour, but it did not complete. I tested with skeletor locally and it could skeletonise them. One suggestion if the file size is too large, would be to simplify the mesh before passing it to skeletor so that the author's implementation can handle it. Another minor point here is that the .obj has to have the mesh's normals encoded for the wavefront algorithm

to work (I think). Many .obj files do not include this data as a mesh can be rendered without it. I tested also with .obj files with their normals stripped. Skeletor throws a warning when it cannot find the normals, but the xyz2swc web-interface does not give any indication as to the issue. The naive user would just assume the tool does not work, and move on. I have attached to this review sample .obj data that I tried and failed to skeletonise with the xyz2swc UI - raw meshes for AOTU19 neurons from the flywire and hemibrain drosophila connectome projects, and simplified versions.

We greatly appreciate the attached mesh data files which allowed us to further test and enhance the capabilities of the xyz2swc software. Specifically, we have now modified the mesh conversion module to (i) automatically simplify large mesh files before conversion (<https://pypi.org/project/fast-simplification/>), and (ii) effortlessly convert mesh files with no encoded normals (by enabling `fix_normals=True` of the `skeletor.pre.fix_mesh()` module). Additionally, we also revised the SWC standardization algorithms, which significantly speeds up the overall conversion process for very large files, e.g., the software is now able to convert ~200MB .obj files in a matter of minutes.

The revised code is now also able to handle any exceptions more gracefully to update the file conversion status to “FAIL”, instead of simply timing-out. For security reasons, we will not print out. Providing stack traces directly is considered a security risk (https://owasp.org/www-project-web-security-testing-guide/stable/4-Web_Application_Security_Testing/08-Testing_for_Error_Handling/01-Testing_For_Improper_Error_Handling), that may be used to gather information about the system for harmful purposes, so therefore we will not provide stack traces to any client application. In future versions, as we gather experience, we will try to identify common error modes, and provide more detailed error messages.

R3.4 I did a bit of debugging and wrote some R functions for API access to xyzswc in my last review and made it available here: <https://github.com/natverse/neuromorphr/blob/master/R/xyz2swc.R>. The authors made corrections to their API and have written their own python script for the same I think, they say in their rebuttal: “we now provide Python code to use the API in the ReadMe file, as recommended, and uploaded the example to GitHub”. However, I cannot find it on their README? I see only a link to the API docs (<https://neuromorpho.org/xyz2swc/docs>). API access will help power users make use of their tool, providing an example means people do not have to figure out its use by themselves.

We thank the reviewer for spotting this discrepancy, an example python client with documentation and unit tests is now available on the GitHub page (in the clients/python folder).

R3.5 After the initial suggestion on synapses, and then a discussion and sample data exchange with the authors by email, the authors now give a brief description of how synapse information can be added to the footer, with specific suggestions on columns (one small update though is to add references 39 and 40 to the description of F2 in line 513, and whatever citations F1 needs on 512).

We have included the citations in the revised manuscript.

R3.6 I think this is a great addition, but would ask the authors to make the minor adjustment of including a supplementary table/figure, similar to their current Table or Figure 1, so as to visually show the suggestion. These visual aids are both helpful and influential in determining what conventions potential users - who may only skim-read the manuscript - eventually adopt. I think doing so would make the suggestion stronger.

Ideally, this would also be done for the metadata suggestion (they could be combined into a single display item to save space). Not least because it was a quibble for 3 out of 4 reviewers.

Thank you for the suggestion. We have now added a new Supplementary Figure S1 which visually shows the recommended optional inclusions: metadata in the header, and synapse connectivity in the footer. The figure is appropriately referenced in the ‘Recommendations for Optional Inclusion of Ancillary Information in SWC Files’ section.

R3.7 I would also recommend including these images on the README for the Github repo and relevant Neuromorpho.org page. This will help the suggested conventions be considered.

We agree, and have now included the new Supplementary Figure S1 on the ReadTheDocs page under a newly created section ‘Recommendations for Optional Inclusion of Ancillary Information’: <https://swc-specification.readthedocs.io/en/latest/swc.html#recommendations-for-optional-inclusion-of-ancillary-information>

R3.8 In summary, the paper should be published. I think the authors might want to consider my suggestions for their error messages/figures/tables/README to help maximise the number of adopters in the next years.

Best wishes and well done!

Alex Bates

Thank you!

Reviewer #4

R4.1 This paper describes a pragmatic and practical approach to standardisation of data for a specific, but large community of researchers. It addresses a clear problem in the domain and demonstrates its effectiveness by converting all existing data and evaluating the results. The application is only applicable for researchers involved in neural morphology work, but this seems to be an active community who would benefit from such standardisation. The authors aim is to adhere to the FAIR principles and much more clearly state how they do that - and also how this work relates to other initiatives in related fields. What is learned from the use of this approach will be applicable for the broader scientific community developing FAIR data solutions.

The authors have addressed my previous comments and concerns (and also many of the concerns of the other reviewers). The manuscript is clearer and has been improved as a result

We are grateful for the positive review.